# Spatiotemporal control of mitotic exit during anaphase by an aurora B-Cdk1 crosstalk

Olga Afonso[1,2†], Colleen M Castellani[3], Liam P Cheeseman[1,2], Jorge G Ferreira[1,2,4], Bernardo Orr[1,2], Luisa T Ferreira[1,2], James J Chambers[5], Eurico Morais-de-Sá[2,6], Thomas J Maresca[3,7], Helder Maiato[1,2,4]*

[1]Chromosome Instability & Dynamics Group, i3S - Instituto de Investigação e Inovação em Saúde, Universidade do Porto, Porto, Portugal; [2]Instituto de Biologia Molecular e Celular, Universidade do Porto, Porto, Portugal; [3]Biology Department, University of Massachusetts, Amherst, United States; [4]Cell Division Group, Experimental Biology Unit, Department of Biomedicine, Faculdade de Medicina, Universidade do Porto, Porto, Portugal; [5]Institute for Applied Life Sciences, University of Massachusetts, Amherst, United States; [6]Epithelial Polarity & Cell Division Group, i3S - Instituto de Investigação e Inovação em Saúde, Universidade do Porto, Porto, Portugal; [7]Molecular and Cellular Biology Graduate Program, University of Massachusetts, Amherst, United States

*For correspondence:
maiato@i3s.up.pt

Present address: [†]Department of Biochemistry, Sciences II, University of Geneva, Geneva, Switzerland

Competing interests: The authors declare that no competing interests exist.

**Abstract** According to the prevailing 'clock' model, chromosome decondensation and nuclear envelope reformation when cells exit mitosis are byproducts of Cdk1 inactivation at the metaphase-anaphase transition, controlled by the spindle assembly checkpoint. However, mitotic exit was recently shown to be a function of chromosome separation during anaphase, assisted by a midzone Aurora B phosphorylation gradient - the 'ruler' model. Here we found that Cdk1 remains active during anaphase due to ongoing APC/C$^{Cdc20}$- and APC/C$^{Cdh1}$-mediated degradation of B-type Cyclins in *Drosophila* and human cells. Failure to degrade B-type Cyclins during anaphase prevented mitotic exit in a Cdk1-dependent manner. Cyclin B1-Cdk1 localized at the spindle midzone in an Aurora B-dependent manner, with incompletely separated chromosomes showing the highest Cdk1 activity. Slowing down anaphase chromosome motion delayed Cyclin B1 degradation and mitotic exit in an Aurora B-dependent manner. Thus, a crosstalk between molecular 'rulers' and 'clocks' licenses mitotic exit only after proper chromosome separation.
DOI: https://doi.org/10.7554/eLife.47646.001

## Introduction

The decision to enter and exit mitosis is critical for genome stability and the control of tissue homeostasis, perturbation of which has been linked to cancer (*Evan and Vousden, 2001*; *Hanahan and Weinberg, 2011*). While the key universal principles that drive eukaryotic cells into mitosis are well established (*Domingo-Sananes et al., 2011*; *Lindqvist et al., 2009*; *Rieder, 2011*), the mechanistic framework that determines mitotic exit remains ill-defined. The prevailing 'clock' model conceives that mitotic exit results from APC/C$^{Cdc20}$-mediated Cyclin B1 degradation when cells enter anaphase, under control of the spindle assembly checkpoint (SAC) (*Musacchio, 2015*). This allows the dephosphorylation of Cdk1 substrates by PP1/PP2A phosphatases, setting the time for chromosome decondensation and nuclear envelope reformation (NER) (*Wurzenberger and Gerlich, 2011*), two hallmarks of mitotic exit. This model is supported by the observation that inhibition of PP1/PP2A or

 

expression of non-degradable Cyclin B1 (and B3 in *Drosophila*) mutants arrested cells in anaphase (*Afonso et al., 2014*; *Parry and O'Farrell, 2001*; *Schmitz et al., 2010*; *Sigrist et al., 1995*; *Vagnarelli et al., 2011*; *Wheatley et al., 1997*; *Wolf et al., 2006*). However, because Cyclin B1 is normally degraded during metaphase (*Clute and Pines, 1999*; *Huang and Raff, 1999*), the data from non-degradable Cyclin B1 expression could be interpreted as a spurious gain of function due to the artificially high levels of Cyclin B1 that preserve Cdk1 activity during anaphase. Whether Cyclin B1-Cdk1 normally plays a role in the control of anaphase duration and mitotic exit remains unknown.

We have recently uncovered a spatial control mechanism that operates after SAC satisfaction to delay chromosome decondensation and normal NER in response to incompletely separated chromosomes during anaphase in *Drosophila* and human cells (*Afonso et al., 2014*). The central player in this mechanism is a constitutive midzone-based Aurora B phosphorylation gradient that was proposed to monitor the position of chromosomes along the spindle axis during anaphase (*Afonso et al., 2014*; *Maiato et al., 2015*). Thus, according to this model, mitotic exit in metazoans, as defined as the irreversible transition into G1 after chromosome decondensation and NER, cannot simply be explained by a 'clock' that starts ticking at the metaphase-anaphase transition, but must also respond to spatial cues as cells progress through anaphase. The main conceptual implication of this 'ruler' model is that mitotic exit is determined during anaphase, and not at the metaphase-anaphase transition under SAC control. In this case, a molecular 'ruler' that prevents precocious chromosome decondensation and NER would allow that all separated sister chromatids end up in two individualized daughter nuclei during a normal mitosis. Moreover, it provides an opportunity for the correction and reintegration of lagging chromosomes that may arise due to deficient interchromosomal compaction in anaphase (*Fonseca et al., 2019*) or erroneous kinetochore-microtubule attachments that are 'invisible' to the SAC (e.g. merotelic attachments) (*Gregan et al., 2011*). Interestingly, Aurora B association with the spindle midzone depends on the kinesin-6/Mklp2/Subito (*Cesario et al., 2006*; *Gruneberg et al., 2004*) and is negatively regulated by Cdk1 (*Hümmer and Mayer, 2009*). Thus, the establishment of a midzone-based Aurora B 'ruler' in anaphase is determined by the sudden drop of Cdk1 activity (the 'clock') at the metaphase-anaphase transition. In the present work, we investigate whether and how molecular 'rulers' also regulate the 'clocks' during anaphase to coordinate mitotic exit in space and time in metazoans.

## Results

### Cyclin B1 continues to be degraded during anaphase and its disappearance is a strong predictor of mitotic exit in metazoans

To investigate a possible role of Cdk1 during anaphase, we started by monitoring Cyclin B1-GFP by spinning-disc confocal microscopy in live *Drosophila* and human cells in culture. Mild induction of Cyclin B1-GFP expression in S2 cells reproduced the localization of endogenous Cyclin B1 in the cytoplasm, mitotic spindle, kinetochores and centrosomes (*Bentley et al., 2007*; *Clute and Pines, 1999*; *Huang and Raff, 1999*; *Pines and Hunter, 1991*), without altering normal anaphase duration or increasing chromosome missegregation (*Figure 1a*, *Figure 1—figure supplement 1a,a'* and *Figure 1—figure supplement 2a–c*). In agreement with previous reports (*Clute and Pines, 1999*; *Huang and Raff, 1999*), cytoplasmic Cyclin B1-GFP levels decreased abruptly at the metaphase-anaphase transition (*Figure 1a,c*). However, in contrast to what has been observed in *Drosophila* embryos (*Huang and Raff, 1999*), the centrosomal pool of Cyclin B1 in S2 cells was the most resistant to degradation and persisted well after anaphase onset, becoming undetectable only ~1 min before DNA decondensation (*Figure 1a,c*, *Figure 1—figure supplement 1a,a'* and *Figure 1—figure supplement 2a,d,e* and *Video 1*). Indeed, complete Cyclin B1 disappearance from centrosomes during anaphase strongly correlated with mitotic exit (*Figure 1—figure supplement 2f*).

To extend the significance of these observations, we monitored endogenously tagged Cyclin B1-Venus levels in human HeLa and hTERT-RPE1 cells (*Collin et al., 2013*) throughout mitosis and found that it also continues to be degraded during anaphase (*Figure 1b,c* and *Figure 1—figure supplement 3a,c* and *Videos 2* and *3*). The physiological relevance of these findings was confirmed in primary adult *Drosophila* follicular epithelium cells (ex vivo) expressing endogenously tagged Cyclin B1-GFP (*Figure 1d,e* and *Video 4*), and after injection of Cyclin B1-mCherry mRNA in mouse oocytes undergoing meiosis II, when chromosomal division is similar to mitosis (*Figure 1f,f'*, *g*).

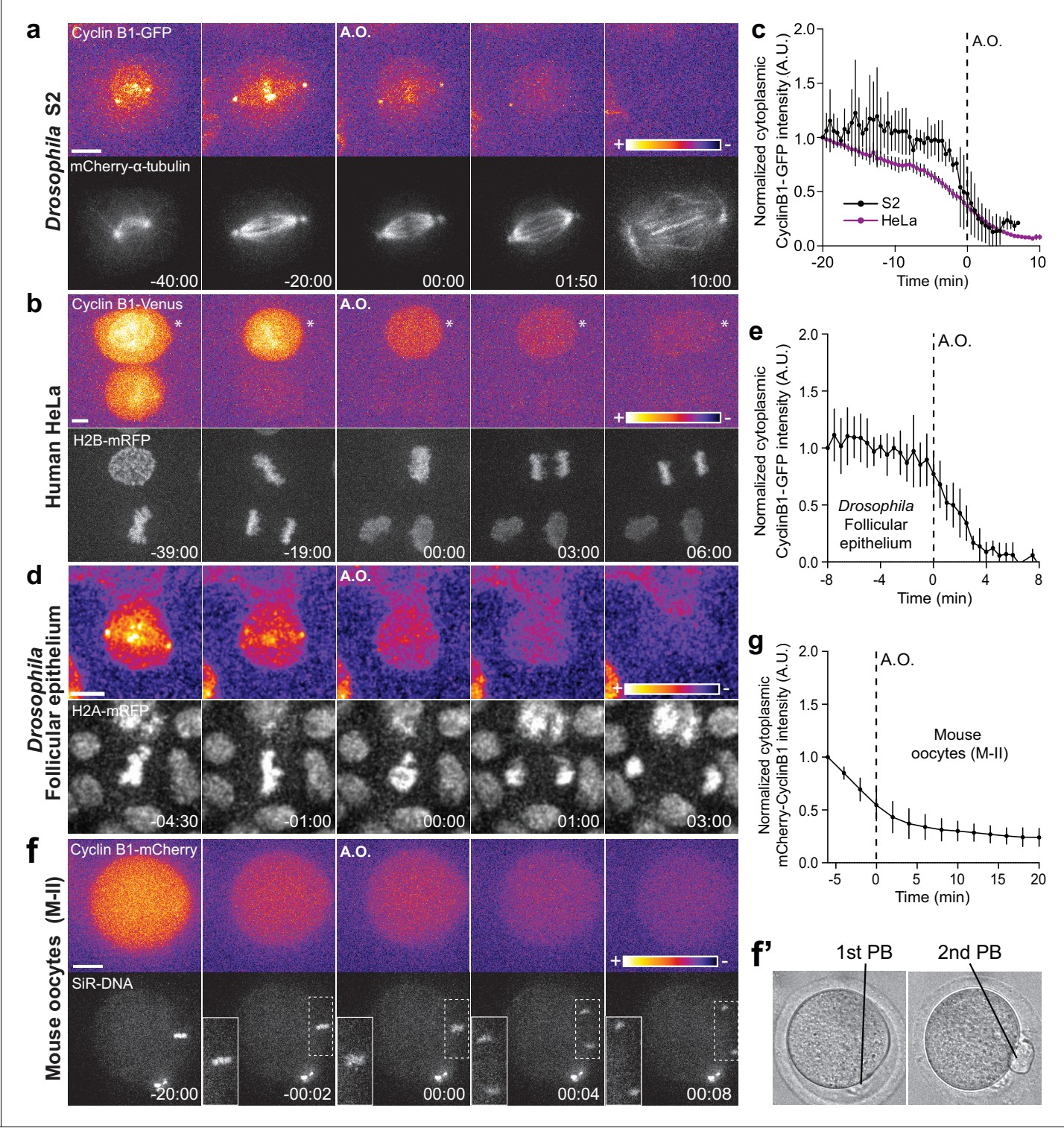

**Figure 1.** Cyclin B1 continues to be degraded during anaphase. (**a**) *Drosophila* S2 cell from nuclear envelope breakdown to NER showing the different pools of Cyclin B1 in the mitotic apparatus. Scale bar is 5 μm. (**b**) Two neighbor HeLa cells (* indicates a cell that is slightly delayed relative to its neighbor; compare relative Cyclin B1 levels between neighbors as they exit mitosis) expressing exogenous H2B-mRFP and showing continuous degradation of endogenous Cyclin B1-Venus during anaphase. Scale bar is 5 μm. (**c**) Cyclin B1 degradation profile in *Drosophila* S2 cells (n = 4 cells) and in HeLa cells with endogenously tagged Cyclin B1-Venus (n = 3 cells). Fluorescence intensity values were normalized to 20 min before anaphase onset (A.O.). (**d**) Time-lapse images of dividing *Drosophila* follicle cells expressing endogenously tagged Cyclin B1-GFP and His2Av-mRFP. Scale bar is 5 μm. (**e**) Quantification of Cyclin B1-GFP fluorescence intensity in the cytoplasm (n = 8 cells, five ovaries). Fluorescence intensity values were

*Figure 1 continued on next page*

*Figure 1 continued*

normalized to 8 min before A.O. (**f**) Time-lapse images of a metaphase II oocyte expressing Cyclin B1-mCherry and stained with SiR-DNA undergoing anaphase II after parthenogenic activation. Inset is 1.5x magnification of separating chromosomes. (**f'**) Images of transmission light microscopy showing the same oocyte prior to and after imaging. Note the presence of the first and second polar bodies. Scale bar is 20 μm. (**g**) Quantification of Cyclin B1-mCherry fluorescence intensity in the cytoplasm (n = 20 oocytes, two independent experiments). Fluorescence intensity values were normalized to 6 min before anaphase onset. The LUT 'fire' is used to highlight Cyclin B1 localization in the different systems. Time in all panels is in min:sec.
DOI: https://doi.org/10.7554/eLife.47646.002

The following source data and figure supplements are available for figure 1:

**Figure supplement 1.** Endogenous Cyclin B1 is detectable on centrosomes in metaphase and anaphase and can localize to spindle midzone microtubules in *Drosophila* S2 cells.
DOI: https://doi.org/10.7554/eLife.47646.003

**Figure supplement 2.** Cyclin B1 degradation during anaphase correlates with DNA decondensation.
DOI: https://doi.org/10.7554/eLife.47646.004

**Figure supplement 3.** Aurora B localization at the spindle midzone impacts Cyclin B1 degradation during anaphase in hTERT-RPE1 cells.
DOI: https://doi.org/10.7554/eLife.47646.005

**Figure supplement 3—source data 1.** Cyclin B1-Venus half-life in control and Mklp2-depleted cells.
DOI: https://doi.org/10.7554/eLife.47646.006

Thus, Cyclin B1 continues to be degraded during anaphase in *Drosophila,* mouse and human cells, including primary tissues, and its disappearance is a strong predictor of mitotic exit.

## Degradation of B-type Cyclins and Cdk1 inactivation during anaphase are rate limiting for mitotic exit

To investigate the functional relevance of an anaphase Cyclin B1 pool we acutely inhibited Cdk1 activity at anaphase onset with RO-3306 (*Vassilev et al., 2006*) and found that it accelerated NER in S2 cells (*Figure 2a–d*). In contrast, expression of non-degradable Cyclin B1 or Cyclin B3 prevented normal spindle elongation and induced an extensive delay in anaphase, in agreement with previous works (*Afonso et al., 2014*; *Parry and O'Farrell, 2001*; *Potapova et al., 2006*; *Sigrist et al., 1995*; *Wolf et al., 2006*). While this delay was dependent on Cdk1 activity (*Figure 2—figure supplement 1a–e*), expression of non-degradable B-type Cyclins does not reflect a physiological role for their degradation in mitotic exit. As so, we also investigated the effect of inhibiting proteasome-mediated degradation of wild-type B-type Cyclins, including endogenously tagged versions, by adding MG132 just before or at anaphase onset in *Drosophila* and human cells, respectively. Under these conditions, cells remained in an anaphase-like state for several hours with separated sister chromatids and detectable Cyclin B1, often reverting to a metaphase-like state (*Figure 3a,d,e*, *Figure 4a,d,e*, Figure 6a,b,f and *Videos 5* and *6*). This 'mitotic reversal' is reminiscent of the one observed in cells treated with MG132 in metaphase after addition and subsequent washout of a Cdk1 inhibitor (*Potapova et al., 2006*).

To evaluate the roles of Cdk1 and Aurora B in mitotic exit, we acutely inhibited either Cdk1 or Aurora B activity in MG132-treated anaphase-like cells. After Aurora B inhibition with Binucleine-2 (*Smurnyy et al., 2010*) or ZM447439 (*Ditchfield et al., 2003*) in *Drosophila* and human cells, respectively, they also remained in an anaphase-like state for several hours and often reverted into a metaphase-like state, but mitotic exit was resumed significantly earlier than MG132-treated controls (*Figure 3b,d*, *Figure 4b,d,e* and *Videos 7* and *8*). In contrast, Cdk1 inhibition immediately triggered chromosome decondensation and mitotic exit

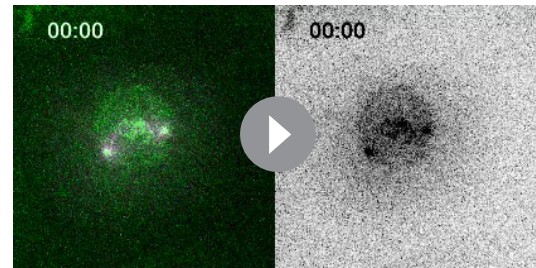

**Video 1.** Exogenous expression of Cyclin B1-GFP in *Drosophila* S2 cells shows continuous degradation during anaphase. Mitotic progression from nuclear envelope breakdown to nuclear envelope reformation in a *Drosophila* S2 cell expressing Cyclin B1-GFP (green) and mCherry-α-tubulin (magenta). Cyclin B1 becomes undetectable at the anaphase-telophase transition, just before mCherry-α-tubulin exclusion from the nucleus. Time is min:sec.
DOI: https://doi.org/10.7554/eLife.47646.007

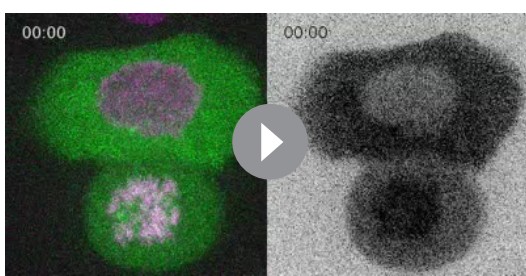
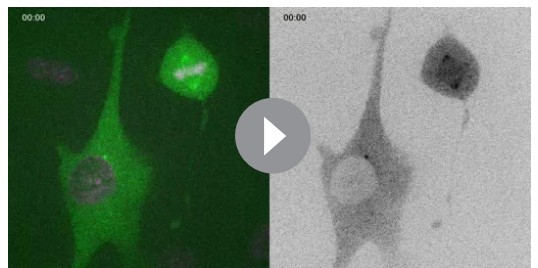

**Video 2.** Endogenous Cyclin B1 is continuously degraded during anaphase in human HeLa cells. Mitotic progression in two slightly asynchronous HeLa cells expressing endogenous Cyclin B1 tagged with Venus (green) and exogenous H2B-mRFP (magenta). Cyclin B1 becomes undetectable at the anaphase-telophase transition, just before DNA decondensation. Time is min:sec.
DOI: https://doi.org/10.7554/eLife.47646.008

**Video 3.** Endogenous Cyclin B1 is continuously degraded during anaphase in human hTERT-RPE1 cells. Mitotic progression of two slightly asynchronous hTERT-RPE1 cells expressing endogenous Cyclin B1 tagged with Venus (green) and SiR-DNA (magenta). Cyclin B1 becomes undetectable at the anaphase-telophase transition, just before DNA decondensation. Time is min:sec.
DOI: https://doi.org/10.7554/eLife.47646.009

(*Figure 3c,d*, *Figure 4c–e* and *Videos 9* and *10*). Thus, Aurora B inhibition is clearly not sufficient to drive cells out of mitosis if Cyclin B1 degradation is prevented, whereas Cdk1 activity during anaphase is rate-limiting for mitotic exit in metazoans.

## Cyclin B1 degradation during anaphase is mediated by APC/C^Cdc20 and APC/C^Cdh1

APC/C^Cdc20 is thought to regulate the metaphase-anaphase transition by targeting Cyclin B1 for degradation through recognition of a D-box (*Glotzer et al., 1991*), under control of the SAC (*Clute and Pines, 1999*). This would then promote APC/C binding to Cdh1 (APC/C^Cdh1), which recognizes substrates with either a D- or a KEN-box (*Lindon, 2008*). However, recent Cryo-electron microscopy studies have established a paradigm of multivalent APC/C-substrate interaction involving two or more degrons, with Cdc20 and Cdh1 having almost identical KEN and D-box receptor sites (*da Fonseca et al., 2011*; *Zhang et al., 2016*). Sequence analysis of *Drosophila* Cyclin B1 revealed a KEN box at position 248, which we mutated to AAN (*Figure 5a–d*). When expressed in S2 cells the KEN-Cyclin B1-GFP mutant was more slowly degraded specifically during anaphase, delaying mitotic exit (*Figure 5a,b,d,f,g*). In agreement, APC/C^Cdh1 depletion in *Drosophila* and human cells was previously shown to cause an accumulation of Cyclin B1 (*Ma et al., 2012*; *Meghini et al., 2016*). To directly test whether APC/C^Cdh1 mediates Cyclin B1 degradation during anaphase we performed RNAi against the *Drosophila* Cdh1 orthologue Fizzy-related (Fzr). As in the case of the KEN-box mutant, Fzr RNAi significantly delayed Cyclin B1 degradation during anaphase, postponing mitotic exit (*Figure 5e–j*). Lastly, we combined our KEN-box mutant with acute APC/C inhibition at anaphase onset with the D-box competitor Apcin (*Sackton et al., 2014*) and found a synergistic effect (*Figure 5— figure supplement 1a–d*). This result suggests the presence of multiple recognition sites that mediate Cyclin B1 degradation through both APC/C^Cdc20 and APC/C^Cdh1 during anaphase.

Although human Cyclin B1 (but not human Cyclin B2) lacks a canonical KEN-box, its recognition by APC/C^Cdh1 might involve a D-box or other motifs, whose inactivation compromises Cyclin B1 degradation in anaphase (*Clijsters et al., 2014*; *Lindon, 2008*;

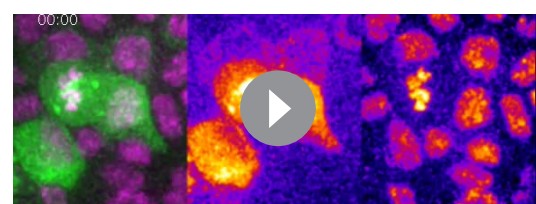

**Video 4.** Endogenous Cyclin B1 is continuously degraded during anaphase in the *Drosophila* adult follicular epithelium. Mitotic progression of dividing follicle cells expressing endogenous Cyclin B1 tagged with GFP (green) and His2Av-mRFP (magenta). Note a pool of cytoplasmic Cyclin B1 that remains detectable until around 4 min after anaphase onset. Time is in min:sec.
DOI: https://doi.org/10.7554/eLife.47646.010

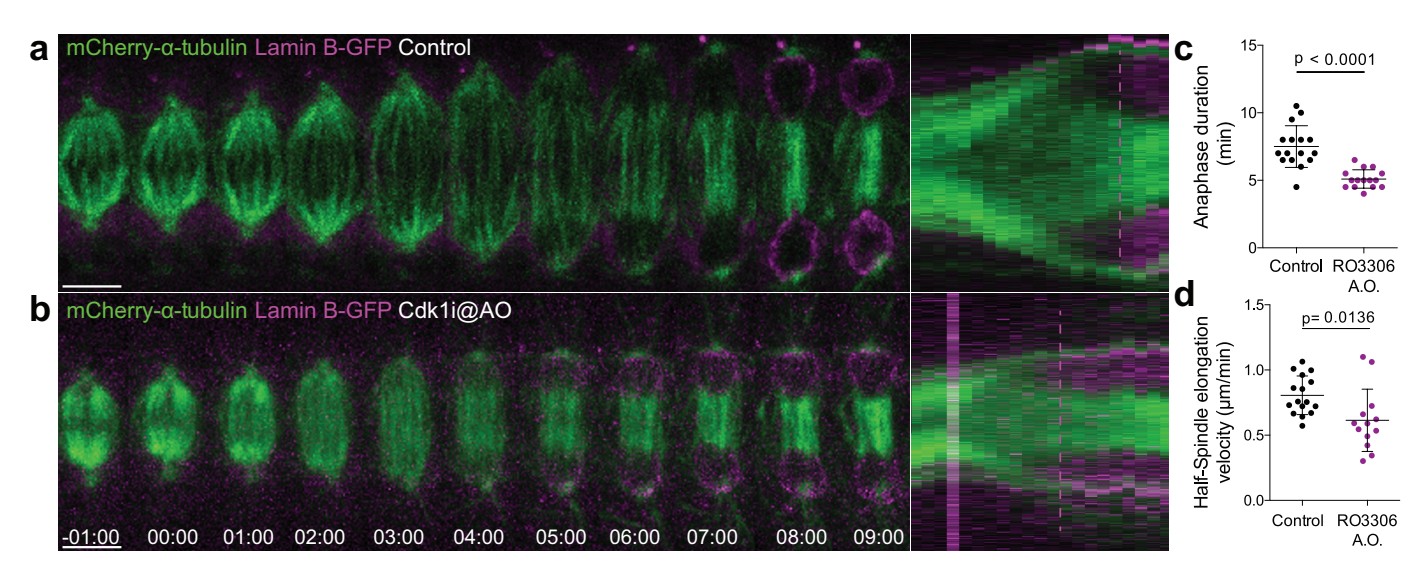

**Figure 2.** Cdk1 inhibition at anaphase onset accelerates NER. (a) and (b) Control and Cdk1-inhibited *Drosophila* S2 cells at anaphase onset (A.O.) stably expressing Lamin B-GFP/mCherry-α-tubulin. Scale bars are 5 µm. Time is in min:sec. Panels on the right side show the corresponding collapsed kymographs. Dashed lines indicate the moment of NER. (c) and (d) Quantification of anaphase duration (control n=16 cells; Cdk1i n=15 cells) and half-spindle elongation velocity (control n=16 cells; Cdk1i, n=13 cells), respectively, in the conditions shown in (a) and (b). Statistical significance was tested with an unpaired t-test.

DOI: https://doi.org/10.7554/eLife.47646.011

The following source data and figure supplements are available for figure 2:

**Source data 1.** Calculation of anaphase duration after Cdk1 inhibition at anaphase onset.
DOI: https://doi.org/10.7554/eLife.47646.014
**Figure supplement 1.** Cdk1 inhibition during anaphase is required for mitotic exit.
DOI: https://doi.org/10.7554/eLife.47646.012
**Figure supplement 1—source data 1.** Calculation of half-spindle elongation velocity in non-degradable Cyclins.
DOI: https://doi.org/10.7554/eLife.47646.013

*Matsusaka et al., 2014*). To investigate whether Cyclin B1 degradation during anaphase is mediated by the APC/C in human cells, we inhibited its activity specifically at anaphase onset with Apcin and/or pro-TAME, a more selective APC/C$^{Cdc20}$ inhibitor (*Sackton et al., 2014*; *Zeng et al., 2010*; *Zhang et al., 2016*) in hTERT-RPE1 cells expressing endogenously tagged Cyclin B1-Venus. Only the simultaneous addition of both drugs at anaphase onset blocked the continuous degradation of Cyclin B1 during anaphase and prevented mitotic exit for several hours, similar to proteasome inhibition (*Figure 6a–f*). Although relative inefficiency of these inhibitors cannot be ruled out, it is conceivable that additive interactions of Cdc20/Cdh1 and Cyclin B1 degrons are required for efficient Cyclin B1 ubiquitination and subsequent degradation by the proteasome. Taken together, these results suggest that *Drosophila* and human Cyclin B1 degradation during anaphase is mediated through the D-box and KEN-box (when present), and can be carried out either by APC/C$^{Cdc20}$ or APC/C$^{Cdh1}$.

### *Drosophila* Cyclin B1 localizes at the spindle midzone in an Aurora B-dependent manner

To investigate the precise localization of Cyclin B1 during anaphase we took advantage of the extremely flat morphology of *Drosophila* S2 cells growing on concanavalin-coated coverslips. This allowed us to detect a faint pool of Cyclin B1-GFP associated with the spindle midzone and midbody during a normal mitosis and cytokinesis, respectively (*Figure 7a,a', b*), as reported previously in *Drosophila* germ cells (*Mathieu et al., 2013*). Detection of the midzone pool of endogenous or ectopically expressed Cyclin B1 was enhanced by abrupt Cdk1 inactivation during metaphase, causing cells to prematurely enter anaphase with high Cyclin B1 levels (*Figure 7c,e; see also Figure 1—figure supplement 1b,b'*). Under these conditions, Cyclin B1 co-localized with Aurora B at the spindle

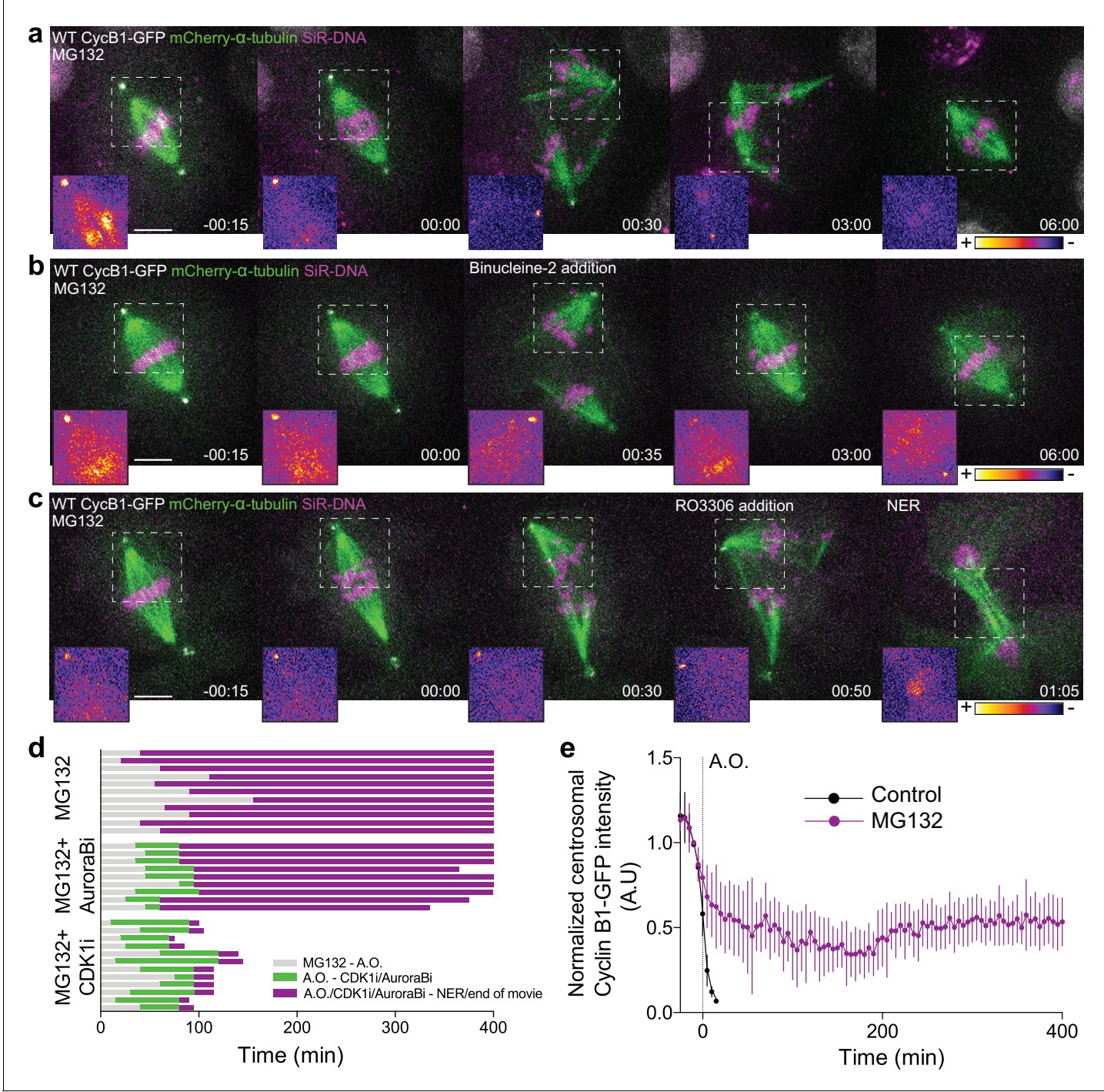

**Figure 3.** Cdk1 inactivation during anaphase licenses mitotic exit and requires proteasome-mediated proteolysis. (a) Representative *Drosophila* S2 cell stably expressing Cyclin B1-GFP/mCherry-α-tubulin and stained with SiR-DNA to follow mitotic chromosomes, showing a strong anaphase arrest after treatment with MG132 (20 μM). Importantly, cells entered anaphase with Cyclin B1 levels compared to untreated control cells, despite the presence of MG132. (b) and (c) *Drosophila* S2 cells arrested in anaphase with MG132, and treated with Aurora B inhibitor (Binucleine-2) or Cdk1 inhibitor (RO3306), respectively, 30–60 min after the anaphase arrest. In (a), (b) and (c) a half-spindle region is highlighted with the LUT 'fire' to reveal Cyclin B1-GFP fluorescence during the anaphase arrest. Scale bars in all panels are 5 μm. Time in all panels is in h:min. (d) Timeline of the experiments shown in (a), (b) and (c) with quantification of total anaphase duration. A.O. = Anaphase onset. Gray color represents the time from MG132 addition to anaphase onset in cases where cells entered anaphase in the presence of MG132. Green color represents the time spent in anaphase until Aurora B or Cdk1 inhibition and purple color represents the time spent in anaphase in the presence of MG132 or MG312+Aurora B inhibition or MG132+Cdk1 inhibition. (e) Cyclin B1-GFP degradation profile in untreated (n = 12 cells) and MG132-treated cells arrested in anaphase (n = 6 cells). A.O. = Anaphase onset.

DOI: https://doi.org/10.7554/eLife.47646.015

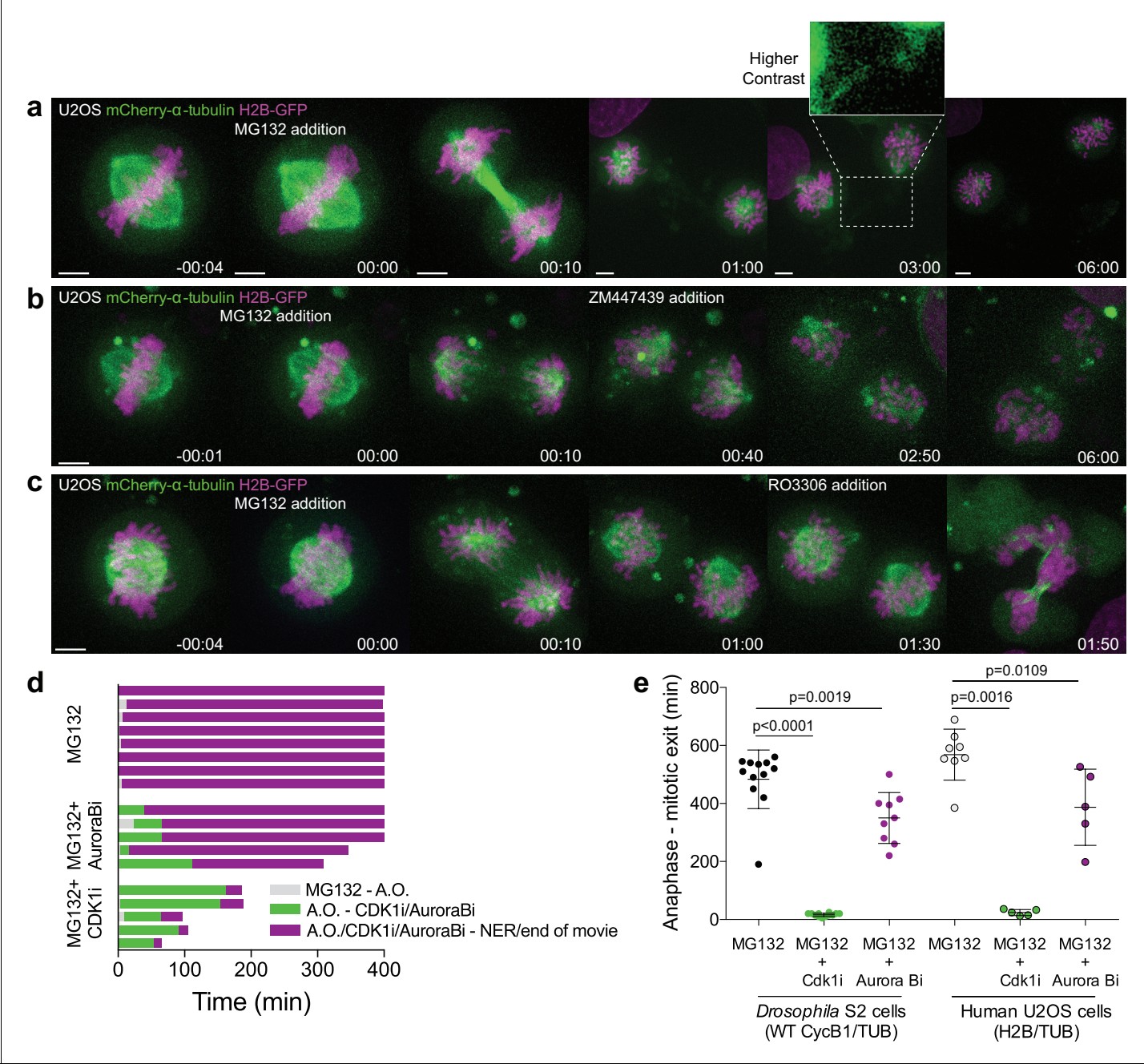

**Figure 4.** Proteasome inhibition at anaphase onset arrests human cells in anaphase in a Cdk1-dependent manner. (**a**) Representative U2OS cell stably expressing H2B-GFP/mCherry-α-tubulin treated with MG132 at anaphase onset (time 00:00). The cell became arrested in anaphase, after full chromosome separation and formation of a spindle midzone. Typically, two new spindles assembled around individual chromatids that erratically attempted to establish new 'metaphase' plates. (**b**) and (**c**) Anaphase arrested U2OS cells obtained by the addition of MG132 were treated with the Aurora B inhibitor (ZM447439) or Cdk1 inhibitor (RO3306), respectively, 30–60 min after anaphase arrest. Time in all panels is in h:min. Scale bars are 5 μm. (**d**) Timeline of the experiments shown in (**a**), (**b**) and (**c**). A.O. = Anaphase onset. Gray color represents the time from MG132 addition to anaphase onset, green color represents the time spent in anaphase until Aurora B or Cdk1 inhibition and purple color represents the time spent in anaphase in the presence of MG132 or MG312+Aurora B inhibition or MG132+Cdk1 inhibition. (**e**) Quantification of total anaphase duration in MG132, MG132 +Aurora B inhibition and MG132+Cdk1 inhibition in *Drosophila* S2 cells and human U2OS cells. Note that for MG132 and MG132+Aurora B inhibition anaphase duration corresponds to the total duration of the movie as most cells do not exit mitosis until the end of acquisition. Statistical significance was tested with a nonparametric Mann-Whitney test.

DOI: https://doi.org/10.7554/eLife.47646.016

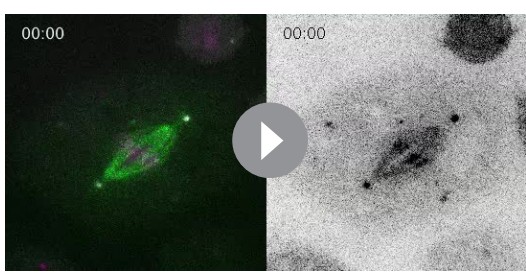

**Video 5.** Proteasome inhibition during anaphase in *Drosophila* cells. *Drosophila* S2 cell stably expressing Cyclin B1-GFP/mCherry-α-tubulin (white/green) and treated with SiR-DNA (magenta) to label mitotic chromosomes. The cell becomes arrested in anaphase for several hours with detectable levels of Cyclin B1 at centrosomes. Time is h:min.
DOI: https://doi.org/10.7554/eLife.47646.017

midzone (*Figure 7—figure supplement 1a–c*). This localization depended on Subito/kinesin-6-mediated midzone targeting and activity of Aurora B (*Figure 7b,d,e*), but was independent of Polo kinase activity (*Figure 7g* and *Figure 7—figure supplement 2*). While we were unable to localize endogenously tagged Cyclin B1-Venus at the spindle midzone in human anaphase cells due to the poor signal/noise at this stage, direct localization of human Cdk1-GFP in live human HeLa cells revealed a clear spindle localization at the metaphase-anaphase transition, with a subsequent enrichment in the central spindle region between separating sister chromosomes throughout anaphase, and midbody during cytokinesis (*Figure 7—figure supplement 3a,b*). In agreement, Cdk1 was recently found enriched on isolated midbodies from human cells (*Capalbo et al., 2019*). These results suggest that Cyclin B1-Cdk1 is spatially regulated by Aurora B activity at the spindle midzone, at least in *Drosophila* cells.

## Aurora B association with the spindle midzone controls mitotic exit by regulating Cdk1 activity during anaphase

We have previously found that Aurora B inhibition at anaphase onset abolished the cellular capacity to delay mitotic exit in response to incomplete chromosome separation during anaphase in both *Drosophila* and human cells (*Afonso et al., 2014*). Here we sought to investigate whether elevating Aurora B protein levels and presumably its activity impacts mitotic exit. For this purpose we overexpressed Aurora B in *Drosophila* S2 cells and found that this delayed its own transition to the spindle midzone and extended anaphase duration (*Figure 7—figure supplement 4a,b,d,e*). Moreover, we

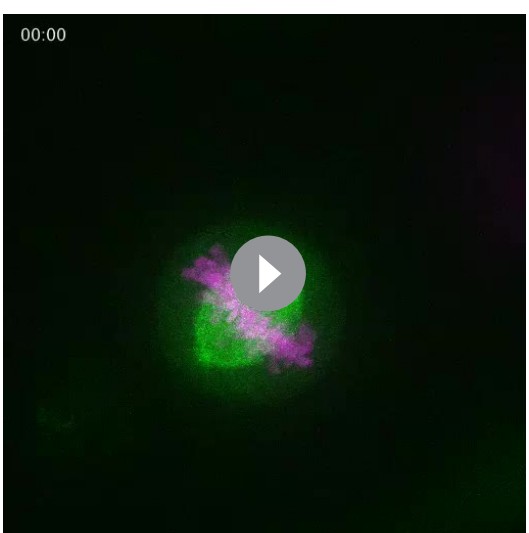

found that Aurora B association with the spindle midzone is a strong predictor of anaphase duration, with several cells overexpressing Aurora B extending anaphase for more than 15 min (*Figure 7—figure supplement 4c,e,f*). Interestingly, this anaphase delay depended on Cdk1 activity

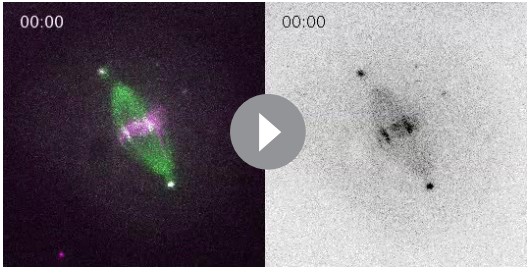

**Video 6.** Proteasome inhibition at anaphase onset arrests human cells in anaphase. Human U2OS cell stably expressing H2B-GFP/mCherry-α-tubulin (magenta/green) treated with MG132 at anaphase onset. Proteasome inhibition induced a strong anaphase arrest for several hours. Time is h:min.
DOI: https://doi.org/10.7554/eLife.47646.018

**Video 7.** Aurora B inhibition in anaphase-arrested *Drosophila* cells treated with MG132. Representative *Drosophila* S2 cell stably expressing Cyclin B1-GFP/mCherry-α-tubulin (white/green) and treated with SiR-DNA (magenta) to label mitotic chromosomes. The Aurora B inhibitor was added 35 min after anaphase onset, however the cell remained arrested in anaphase for several hours. Time is h:min.
DOI: https://doi.org/10.7554/eLife.47646.019

(*Figure 7—figure supplement 4c*), suggesting that Aurora B association with the spindle midzone is able to delay mitotic exit by regulating Cdk1 activity during anaphase. To test this, we monitored Cyclin B1-GFP/Venus degradation after Subito/Mklp2/kinesin-6 RNAi in *Drosophila* S2 and human hTERT-RPE1 cells. This caused a short, but significant delay in Cyclin B1-GFP/Venus degradation during anaphase, increasing Cyclin B1 half-life and often delaying mitotic exit (*Figure 8a–e*; see also *Figure 1—figure supplement 3a–f*). Overall, these data support a model in which the establishment of an anaphase Aurora B phosphorylation gradient concentrates Cyclin B1 and consequently Cdk1 activity at the spindle midzone.

## Incompletely separated chromosomes positioned near the spindle midzone show the highest Cdk1 activity during anaphase

To directly assess the spatiotemporal properties of Cdk1 activity throughout the cell cycle we next employed Fluorescence Lifetime Imaging Microscopy (FLIM) of an established Fluorescence Resonance Energy Transfer (FRET)-based biosensor for Cyclin B1-Cdk1 activity (*Gavet and Pines, 2010*) that was targeted to chromatin in

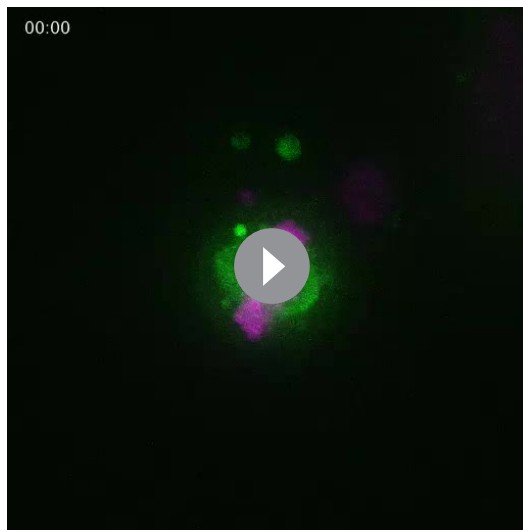

**Video 8.** Aurora B inhibition in anaphase-arrested human cells treated with MG132. Human U2OS cell stably expressing H2B-GFP/mCherry-α-tubulin (magenta/green) treated with MG132 at anaphase onset. The Aurora B inhibitor was added 40 min after anaphase onset. Similarly to the results in *Drosophila* S2 cells, U2OS cells remained arrested in anaphase for several hours after Aurora B inhibition. Time is h:min.
DOI: https://doi.org/10.7554/eLife.47646.020

*Drosophila* S2 cells by fusing it to the C-terminus of human histone H2B. The reporter was modified to make it amenable to FLIM by replacing the donor (mCerulean) with mTurquoise2, which exhibits a mono-exponential lifetime, and acceptor (YPet) with mVenus. The FRET pairs flank the auto-phosphorylation site of human Cyclin B1 and the Polo Box Domain (PBD) of Polo-like kinase one such that it undergoes a conformational change when it is phosphorylated by Cdk1 that yields increased FRET in mitosis and decreased FRET in interphase. Importantly, the biosensor was shown to be highly specific for Cyclin B1-Cdk1 and non-responsive to activities of the midzone-enriched kinases Aurora B and Plk1 (*Gavet and Pines, 2010*). A non-phosphorylatable chromatin-targeted control

with alanine substitutions in the substrate sequence was used to confirm that measured changes in donor lifetime and FRET efficiency were due specifically to phosphorylation rather than non-specific changes in conformation between interphase and mitosis. There was not a statistically significant difference in any of the measured donor lifetime parameters (average lifetime, short lifetime, % short lifetime, % long lifetime) between the non-phosphorylatable control sensor in interphase and mitosis or between the non-phosphorylatable control sensor (interphase and mitosis) and the phosphorylatable sensor in interphase (*Figure 9—source data 1*). Importantly, there were statistically significant differences between the lifetime parameters of the chromatin-targeted phosphorylatable reporter in mitosis versus interphase, most notably the mean donor lifetime decreased from 3.43 ns +/- 0.12 ns (mean +/- S.D.) in interphase

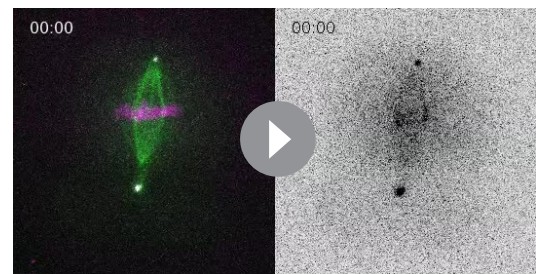

**Video 9.** Cdk1 inhibition in anaphase-arrested *Drosophila* cells treated with MG132. Representative *Drosophila* S2 cell stably expressing Cyclin B1-GFP/mCherry-α-tubulin (white/green) and treated with SiR-DNA (magenta) to label mitotic chromosomes. The Cdk1 inhibitor was added 50 min after anaphase onset, inducing an almost immediate mitotic exit. Time is h:min.
DOI: https://doi.org/10.7554/eLife.47646.021

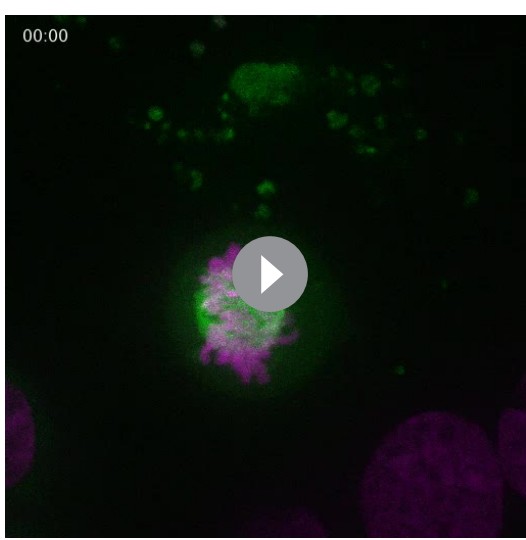

**Video 10.** Cdk1 inhibition in anaphase-arrested human cells treated with MG132. Human U2OS cell stably expressing H2B-GFP/mCherry-α-tubulin (magenta/green) treated with MG132 at anaphase onset. The Cdk1 inhibitor was added 90 min after anaphase onset and induced immediate mitotic exit. Time is h:min.
DOI: https://doi.org/10.7554/eLife.47646.022

to 3.15 ns + /- 0.28 ns in mitosis (includes prophase, prometaphase, and metaphase cells) (*Figure 9a*, *Figure 9—source data 1*). The measured decrease in donor lifetime of the sensor on mitotic chromatin is indicative of increased FRET during mitosis. Accordingly, there was a statistically significant increase in the FRET efficiency (*E*) of the phosphorylatable sensor in mitosis (11.78% + /- 7.89%) relative to interphase (4.12% + /- 3.25%), which was statistically indistinguishable (p>0.05) from the non-phosphorylatable reporter in either interphase or mitosis (*Figure 9a*, *Figure 9—source data 2*). Thus, the chromatin bound reporter specifically reports on elevated Cyclin B1-Cdk1 activity in mitosis relative to interphase.

Next, we performed FLIM-FRET imaging of the phosphorylatable chromatin-associated reporter in S2 cells with spontaneous lagging chromosomes to measure the spatial properties of CyclinB1-Cdk1 activity during anaphase. Remarkably, there was a statistically significant reduction in the donor lifetime (2.61 ns + /- 0.43 ns) for the sensor on lagging chromosomes in the vicinity of the spindle midzone compared to the donor lifetime (3.16 ns + /- 0.20 ns) on the segregated chromosomes (*Figure 9a,b*). This corresponds to ~3 fold increase in the FRET *E* of the sensor on lagging chromosomes (27.0% + /- 12.0%) relative to the FRET *E* of the reporter on segregated DNA (11.7% + /- 5.5%) (*Figure 9a,c*). The FLIM-FRET data support the conclusion that there is a gradient of CyclinB1-Cdk1 activity in the vicinity of the midzone during anaphase that locally phosphorylates chromatin bound substrates on incompletely separated chromosomes. We argue that this activity gradient, which is reminiscent of a previously visualized midzone-based gradient of Aurora B kinase activity (*Afonso et al., 2014*; *Fuller et al., 2008*), specifically reports on midzone-proximal Cyclin B1-Cdk1 activity rather than that of other midzone-enriched kinases since the sensor is insensitive to inhibition of Aurora B or Plk1 activities (*Gavet and Pines, 2010*) and the activity gradient was no longer evident on lagging chromosomes/chromatin following addition of the Cdk1 inhibitor RO-3306 (*Figure 9A–C*).

## Experimental reduction of anaphase chromosome motion delays Cyclin B1 degradation and mitotic exit in an Aurora B-dependent manner

Aurora B has been shown to mediate Cyclin B1 phosphorylation in *Drosophila* germline and human mitotic cells (*Kettenbach et al., 2011*; *Mathieu et al., 2013*). As so, we investigated whether Aurora B activity is required for Cyclin B1 degradation during anaphase and consequently mitotic exit in both *Drosophila* and human cells. First, we monitored Cyclin B1-GFP/Venus levels after acute inhibition of Aurora B activity at anaphase onset by treating S2 and hTERT-RPE1 cells with Binucleine-2 or ZM447439, respectively. We found that inhibition of Aurora B activity did not accelerate Cyclin B1 degradation during anaphase (*Figure 10—figure supplement 1a,b*). Likewise, acute PP1/PP2A phosphatase inhibition with okadaic acid at anaphase onset did not interfere with normal Cyclin B1 degradation during anaphase (*Figure 10—figure supplement 1c*). These results are consistent with our previous experiments, which showed that Aurora B inhibition at anaphase onset did not accelerate mitotic exit in both *Drosophila* and human cells (*Afonso et al., 2014*). In addition, these data indicate that PP1/PP2A phosphatase activity is not required to regulate Cyclin B1 degradation, namely its centrosomal-associated pool, during anaphase.

In light of the results obtained with our chromosome-targeted Cdk1 FRET sensor, we decided to investigate whether Cyclin B1 degradation during anaphase is a function of chromosome separation. To do so, we slowed down chromosome separation by allowing S2 cells to enter anaphase after a

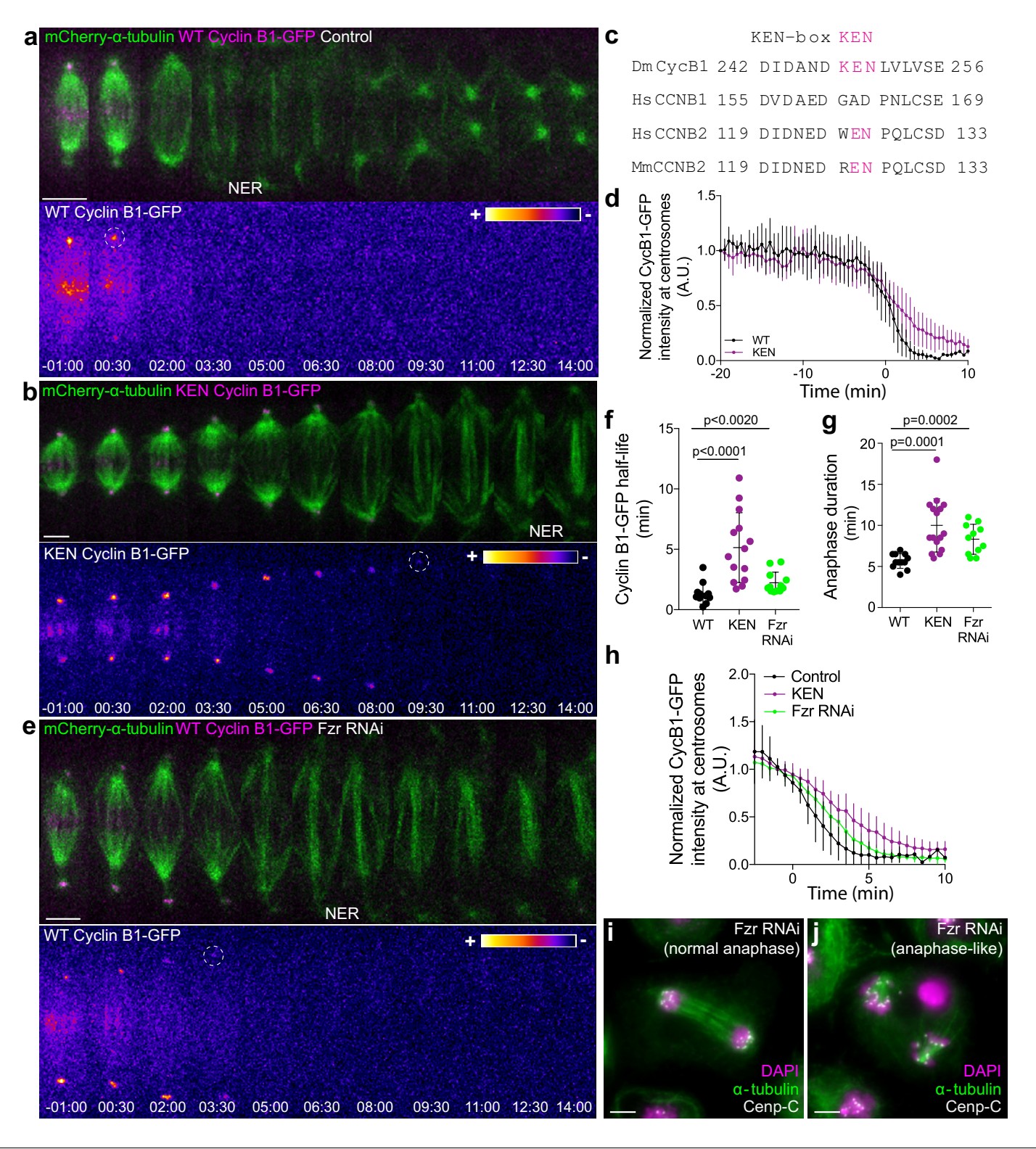

**Figure 5.** APC/C^Cdc20 and APC/C^Cdh1 are required for Cyclin B1 degradation during anaphase and timely mitotic exit. (a) and (b) *Drosophila* S2 cells stably expressing WT Cyclin B1 or a KEN-box mutant version co-expressing mCherry-α-tubulin. (c) Sequence alignment showing the conservation of the *Drosophila* Cyclin B1 KEN-box with mammalian Cyclin B2, but not Cyclin B1. (d) Degradation profile of Cyclin B1-GFP quantified by measuring the GFP fluorescence intensity at centrosomes in control (n = 6 cells) and KEN-box mutant cells (n = 10 cells). Note that KEN-box Cyclin B1 degradation is only

*Figure 5 continued on next page*

*Figure 5 continued*

affected during anaphase. Anaphase onset = 0 min. (**e**) *Drosophila* S2 cell depleted of Fzr and expressing WT CyclinB1-GFP/mCherry-α-tubulin. Cyclin B1-GFP signal is highlighted with the LUT 'fire' and dashed white circles highlight the frame before Cyclin B1 signal disappearance from centrosomes. Scale bars are 5 μm. Time in all panels is in min:sec. (**f**) Quantification of Cyclin B1 half-life (0–4.5 min after anaphase onset) duration in control (n=11 cells), KEN-box mutant (n=12 cells) and Fzr-depleted cells (n=11 cells, pooled from 3 independent experiments) and (**g**) anaphase duration in control (n = 11 cells), KEN-box mutant (n = 16 cells) and Fzr-depleted cells (n = 12 cells, pooled from three independent experiments). Statistically significant differences for anaphase duration and CyclinB1 half-life were tested with an unpaired t-test and a nonparametric Mann-Whitney test, respectively. (**h**) Degradation profile of Cyclin B1-GFP quantified by measuring the GFP fluorescence intensity at centrosomes in control (n = 12 cells), KEN-box mutant cells (n = 14 cells) and Fzr-depleted cells (n = 12 cells). Anaphase onset = 0 min. (**i**) and (**j**) Images of anaphase *Drosophila* S2 cells after Fzr RNAi, fixed and co-stained with Cenp-C and α-tubulin. Note that amongst apparently normal anaphase cells (**i**), anaphase-like cells with clearly separated sister chromatids attached to two half-spindles could also be identified (**j**). Scale bars are 5 μm.

DOI: https://doi.org/10.7554/eLife.47646.023

The following source data and figure supplements are available for figure 5:

**Source data 1.** Anaphase duration after KEN-box mutation or FZR RNAi.

DOI: https://doi.org/10.7554/eLife.47646.026

**Figure supplement 1.** APC/C inhibition at anaphase onset shows a synergistic effect with expression of KEN-box Cyclin B1 mutant.

DOI: https://doi.org/10.7554/eLife.47646.024

**Figure supplement 1—source data 1.** Cyclin B1- GFP half-life after KEN-box mutation and/or APCin treatment at anaphase onset.

DOI: https://doi.org/10.7554/eLife.47646.025

short exposure (less than 15 min) to 10 nM taxol or after depletion of the kinesin-13 KLP10A by RNAi (*Afonso et al., 2014*; *Rogers et al., 2004*). We found that, under both conditions, Cyclin B1-GFP degradation during anaphase was significantly delayed (*Figure 10a–c,e–g*), consistent with the previously reported delay in mitotic exit (*Afonso et al., 2014*). To test whether the observed delay in Cyclin B1 degradation during anaphase was dependent on Aurora B activity, we acutely inhibited Aurora B with Binucleine-2 specifically at anaphase onset in taxol-treated cells. We found that Aurora B inhibition fully restored normal Cyclin B1 degradation in cells with slower chromosome separation (*Figure 10d,f,g*). Importantly, the short exposure of cells to 10 nM taxol did not prevent normal Aurora B accumulation at the spindle midzone (*Figure 10—figure supplement 2a–c*). Overall, these data link chromosome separation to Aurora B-mediated control of Cyclin B1 degradation during anaphase that ultimately licenses mitotic exit.

## Discussion

Taken together, our work reveals that degradation of B-type Cyclins specifically during anaphase is rate-limiting for mitotic exit among animals that diverged more than 900 million years ago. Most importantly, we show that Cdk1 activity during anaphase is a function of chromosome separation and is spatially regulated by Aurora B localization and activity at the spindle midzone. In concert with previous work (*Afonso et al., 2014*), our findings unveil an unexpected crosstalk between molecular 'rulers' (Aurora B) and 'clocks' (B-type Cyclins-Cdk1) that ensures that cells only exit mitosis after proper chromosome separation during anaphase, consistent with the previously proposed chromosome separation checkpoint hypothesis (*Afonso et al., 2014*; *Maiato et al., 2015*). An Aurora B-dependent spatial control mechanism regulating normal NER in human cells has been recently confirmed (*Liu et al., 2018*). However, nuclear envelope defects associated with incomplete chromosome separation during anaphase (namely, anaphase lagging chromosomes due to mitotic errors) were proposed as an inevitable pathological condition. The present work provides yet additional evidence for a molecular network operating during anaphase that promotes chromosome segregation fidelity by controlling mitotic exit in space and time (*Figure 11*). According to this model, APC/C^Cdc20 mediates the initial degradation of Cyclin B1 during metaphase under SAC control. The consequent decrease in Cdk1 activity as cells enter anaphase targets Aurora B to the spindle midzone (via Subito/Mklp2/kinesin-6); Aurora B at the spindle midzone (counteracted by PP1/PP2A phosphatases on chromatin [*Vagnarelli et al., 2011*]) establishes a phosphorylation gradient that locally delays APC/C^Cdc20- and APC/C^Cdh1-mediated degradation of residual Cyclin B1 (and possibly B3) at the spindle midzone, at least in *Drosophila* cells. Localization experiments in human cells suggest that Cdk1 itself might be enriched at the spindle midzone. Consequently, as chromosomes

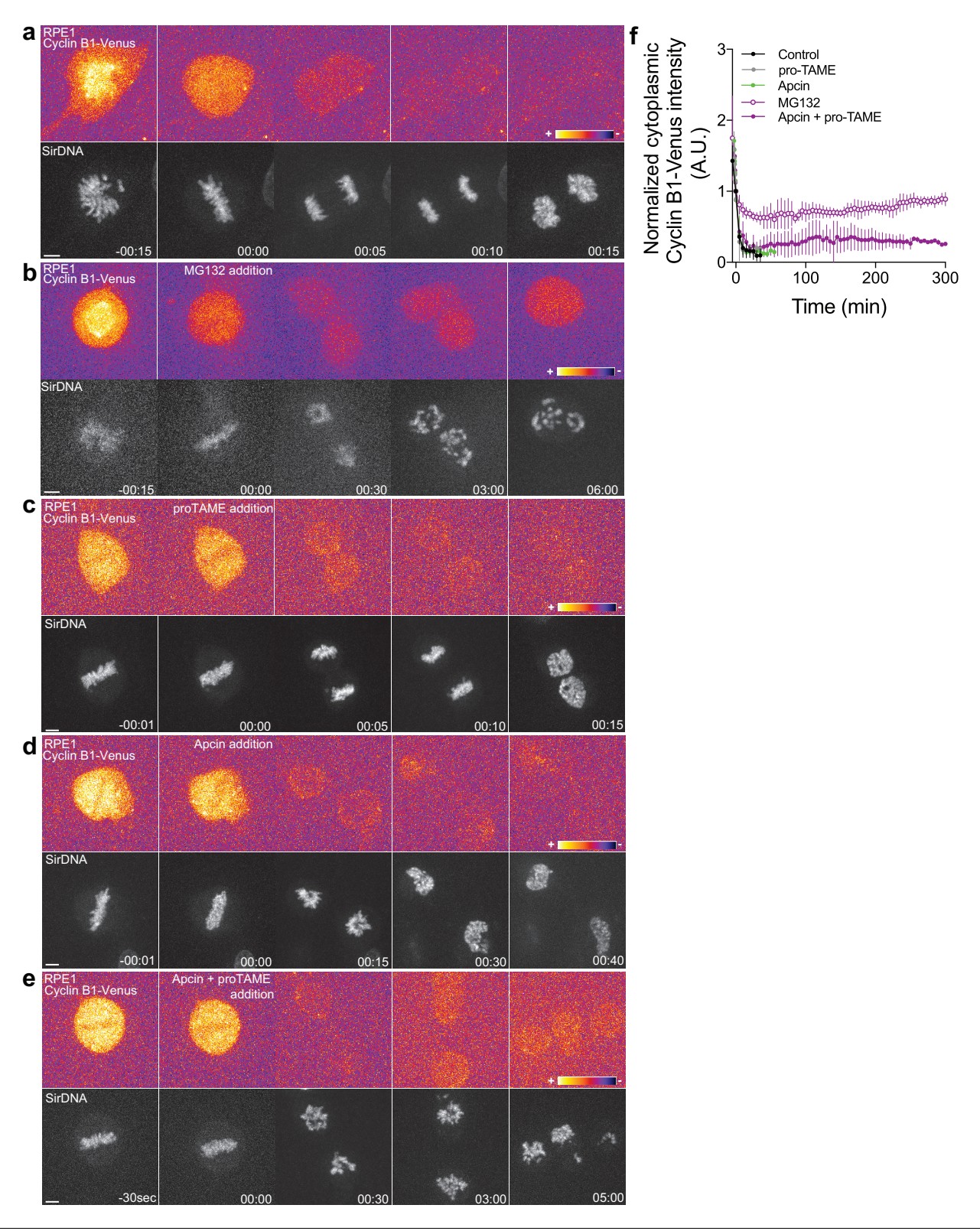

**Figure 6.** Proteasome and APC/C inhibition at anaphase onset induces an anaphase arrest in human hTERT-RPE1 cells. (**a**) Control hTERT-RPE1 cell expressing endogenous Cyclin B1-Venus and co-stained with SiR-DNA to visualize mitotic chromosomes. (**b**) MG132 addition just before or at anaphase onset (time 00:00), caused an anaphase arrest with detectable Cyclin B1 levels and condensed chromosomes. The arrest was sustained up to 6 hr (time window of acquisition). (**c**) pro-TAME addition at anaphase onset (time 00:00) showed no visible effect during anaphase. (**d**) Apcin addition at anaphase

*Figure 6 continued on next page*

*Figure 6 continued*

onset (time 00:00) induced a slight delay in DNA decondensation. (**e**) APC/C inhibition with a cocktail of pro-TAME and Apcin at anaphase onset (time 00:00) caused a strong anaphase delay (during the time window of acquisition – 5 hr). Cyclin B1 localization is highlighted with LUT 'fire'. Scale bars are 5 μm. Time is in h:min. (**f**) Quantification of Cyclin B1-Venus (normalized to anaphase onset) in control (n = 7 cells), MG132 (n = 4 cells), pro-TAME (n = 7 cells), Apcin (n = 6 cells) and pro-TAME+Apcin (n = 4 cells).
DOI: https://doi.org/10.7554/eLife.47646.027

separate and move away from the spindle midzone, Cdk1 activity decreases, allowing the PP1/PP2A-mediated dephosphorylation of Cdk1 and Aurora B substrates (e.g. Lamin B and Condensin I) necessary for mitotic exit. This model is consistent with the recent demonstration that Cdk1 inactivation promotes the recruitment of PP1 phosphatase to chromosomes to locally oppose Aurora B phosphorylation (*Qian et al., 2015*) and recent findings in budding yeast demonstrating equivalent phosphorylation and dephosphorylation events during mitotic exit (*Touati et al., 2018*). It is also consistent with a premature Greatwall inactivation and PP2A:B55 reactivation that would be predicted after acute Cdk1 inactivation during anaphase (*Cundell et al., 2013*; *Cundell et al., 2016*). Most important, this model provides an explanation for the coordinated action of two unrelated protein kinases that likely regulate multiple substrates required for mitotic exit (*Afonso et al., 2017*; *Kettenbach et al., 2011*; *Petrone et al., 2016*).

Previous landmark work has carefully monitored the kinetics of Cyclin B1 degradation in living human HeLa and rat kangaroo Ptk1 cells during mitosis, and concluded that Cyclin B1 was degraded by the end of metaphase, becoming essentially undetectable as cells entered anaphase (*Clute and Pines, 1999*). However, we noticed that, consistent with our findings, a small pool of Cyclin B1 continued to be degraded during anaphase in Ptk1 cells (*Clute and Pines, 1999*). Subsequent work investigating cellular response to anti-mitotic drugs has also shown that human DLD-1 cells undergoing normal mitosis entered anaphase with as much as 32% of Cyclin B1 compared to metaphase levels (*Gascoigne and Taylor, 2008*), suggesting that human cells enter anaphase with significant Cdk1 activity. Indeed, quantitative analysis with a FRET biosensor in human HeLa cells also revealed residual Cdk1 activity during anaphase (*Gavet and Pines, 2010*). However, the significance of persistent Cdk1 activity for the control of anaphase duration and mitotic exit was not investigated in these original studies. Previous works also clearly demonstrated that forcing Cdk1 activity during anaphase through expression of non-degradable Cyclin B1 (and Cyclin B3 in *Drosophila*) prevents chromosome decondensation and NER (*Parry and O'Farrell, 2001*; *Wheatley et al., 1997*; *Wolf et al., 2006*). However, while these works suggested the existence of different Cyclin B1 thresholds that regulate distinct mitotic transitions, expression of non-degradable Cyclin B1 could be interpreted as an artificial gain of function that preserves Cdk1 activity during anaphase. For example, it was shown that expression of non-degradable Cyclin B1 during anaphase 'reactivates' the SAC, inhibiting APC/C$^{Cdc20}$ (*Clijsters et al., 2014*; *Rattani et al., 2014*; *Vázquez-Novelle et al., 2014*). Our work demonstrates in five different experimental systems, from flies to humans, including primary tissues, that Cdk1 activity persists during anaphase and is rate-limiting for the control of mitotic exit. Failure to degrade B-type Cyclins during anaphase blocked cells in an anaphase-like state with separated sister chromatids that remained condensed for several hours, whereas complete Cdk1 inactivation in anaphase triggered chromosome decondensation and NER. Importantly, if a positive feedback loop imposed by phosphatases was sufficient to drive mitotic exit simply by reverting the effect of Cdk1 phosphorylation prior to anaphase, cells would exit mitosis regardless of the remaining pool of B-type Cyclins that sustains Cdk1 activity during anaphase. The main conceptual implication of these findings is that, contrary to what was previously assumed, mitotic exit is determined during anaphase and not at the metaphase-anaphase transition under SAC control.

Our model also implies that persistent Cyclin B1-Cdk1 in anaphase is spatially regulated by a midzone Aurora B gradient. Indeed, we were able to identify a residual pool of Cyclin B1-Cdk1 enriched at the spindle midzone and midbody and showed that this localization was dependent on Aurora B activity and localization at the spindle midzone. Interestingly, human Cyclin B2 (which contains a recognizable KEN box), as well as Cdk1, were identified at the midbody and Cdk1 inactivation during late mitosis was required for the timely completion of cytokinesis in human cells (*Capalbo et al., 2019*; *Mathieu et al., 2013*). Thus, it is possible that in human cells, Cdk1 activity during anaphase is regulated not only by Cyclin B1, but also by Cyclin B2. Importantly, this model predicted the

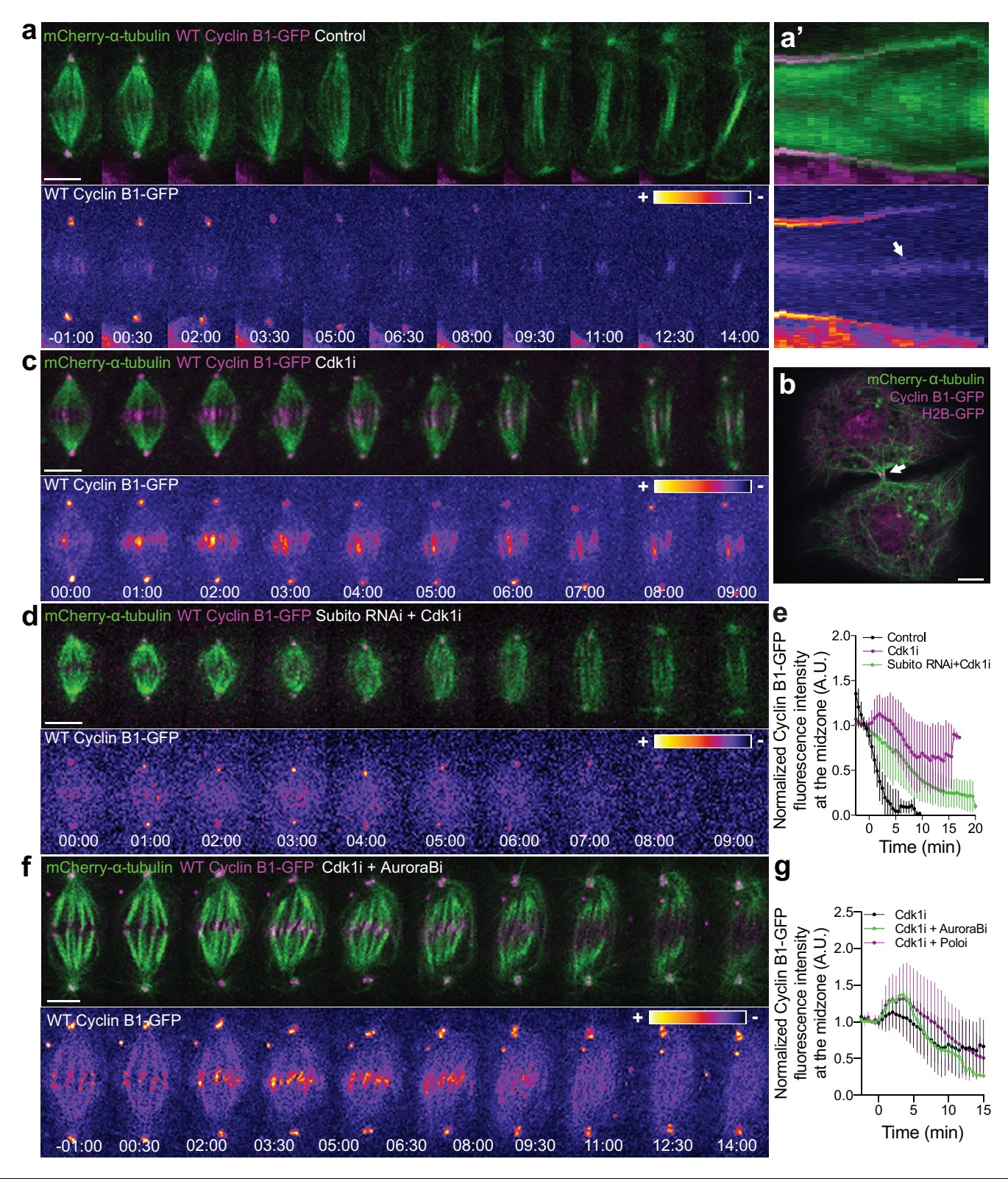

**Figure 7.** Cyclin B1 localization at the spindle midzone depends on Aurora B localization and activity during anaphase. (a) Control *Drosophila* S2 cell stably expressing Cyclin B1-GFP/mCherry-α-tubulin showing a faint pool of Cyclin B1-GFP at the spindle midzone/midbody. (a') Collapsed kymograph of the cell in a where Cyclin B1-GFP can be visualized at the spindle midzone/midbody. (b) Snapshot of a live *Drosophila* S2 cell expressing Cyclin B1-GFP/mCherry-α-tubulin where a midbody pool of Cyclin B1-GFP can be detected before completion of cytokinesis. (c) *Drosophila* S2 cell treated with

*Figure 7 continued on next page*

*Figure 7 continued*

Cdk1 inhibitor during metaphase. Cyclin B1 is not fully degraded and becomes visibly associated with midzone microtubules as cells are forced to exit mitosis. (d) Cdk1 inhibition at metaphase in a *Drosophila* S2 cell expressing Cyclin B1-GFP/mCherry-α-tubulin after Subito/Mklp2 depletion by RNAi. The Cyclin B1 midzone localization is no longer detectable. (e) Quantification of Cyclin B1-GFP fluorescence intensity measured at the spindle midzone (identified by the mCherry-α-tubulin signal) in untreated (n = 11 cells), Cdk1 inhibited cells (n = 10 cells) and Cdk1 inhibition after Subito/Mklp2 depletion (n = 12 cells, pooled from two independent experiments). (f) *Drosophila* S2 cells treated with Cdk1 inhibitor during metaphase and Aurora B inhibitor 4 min after Cdk1 inhibition. For all conditions, Cyclin B1-GFP signal is highlighted with the LUT 'fire'. Scale bar is 5 μm. (g) Quantification of Cyclin B1-GFP fluorescence intensity measured at the spindle midzone in Cdk1 inhibited cells (n = 10 cells), Cdk1 + Aurora B inhibition (n = 11 cells) and Cdk1 + Polo inhibition (n = 10 cells).

DOI: https://doi.org/10.7554/eLife.47646.028

The following source data and figure supplements are available for figure 7:

**Figure supplement 1.** Cyclin B1 co-localization with Aurora B at the spindle midzone after Cdk1 inhibition in metaphase.

DOI: https://doi.org/10.7554/eLife.47646.029

**Figure supplement 2.** Cyclin B1 localization with midzone microtubules is not dependent on Polo kinase activity.

DOI: https://doi.org/10.7554/eLife.47646.030

**Figure supplement 3.** Cdk1 is enriched in the central spindle and midbody in human cells.

DOI: https://doi.org/10.7554/eLife.47646.031

**Figure supplement 4.** Aurora B overexpression induces a Cdk1-dependent anaphase delay.

DOI: https://doi.org/10.7554/eLife.47646.032

**Figure supplement 4—source data 1.** Anaphase duration after Aurora B overexpression.

DOI: https://doi.org/10.7554/eLife.47646.033

existence of a midzone-centered Cdk1 activity gradient during anaphase, which we confirmed experimentally by targeting a FRET reporter of Cdk1 activity to chromosomes.

Finally, our experiments indicate that Aurora B activity regulates Cyclin B1 homeostasis and consequently anaphase duration in the presence of incompletely separated chromosomes. One possibility is that direct Cyclin B1 phosphorylation by Aurora B (*Kettenbach et al., 2011*; *Mathieu et al., 2013*) spatially regulates Cyclin B1 degradation during anaphase, mediated by both APC/C$^{Cdc20}$ and APC/C$^{Cdh1}$. Another non-mutually exclusive possibility is that Aurora B indirectly controls Cyclin B1 during anaphase by regulating APC/C$^{Cdc20}$ and/or APC/C$^{Cdh1}$ activity, as recently shown for Cdk1 (*Fujimitsu et al., 2016*; *Zhang et al., 2016*). Future work will be necessary to test these hypotheses.

In conclusion, we uncovered an unexpected level of regulation at the end of mitosis in metazoans and reconciled what were thought to be antagonistic models of mitotic exit relying either on molecular 'clocks' or on 'rulers'. These findings have profound implications to our fundamental understanding of how tissue homeostasis is regulated, perturbation of which is a hallmark of human cancers.

## Materials and methods

**Key resources table**

| Reagent type (species) or resource | Designation | Source or reference | Identifiers | Additional information |
|---|---|---|---|---|
| Genetic reagent (*Drosophila*) | CycB1-GFP | *Buszczak et al., 2007* | | genotype: w; P{PTT-GC}ycB$^{CC01846}$; P{His2Av-mRFP}/+ |
| Biological sample (mice) | Oocytes from CD-1 mice | Charles River Laboratories | RRID:MGI:5652464 | |
| Cell line (*Drosophila*) | S2-U | Gohta Goshima | | |
| Cell line (*Drosophila*) | S2 H2B-GFP/mCherry-α-tubulin | *Afonso et al., 2014* | | |
| Cell line (*Drosophila*) | S2 Lamin B-GFP/mCherry-α-tubulin | *Afonso et al., 2014* | | |
| Cell line (*Drosophila*) | S2 KEN-Cyclin B1-GFP/mCherry-α-tubulin | This work | | |

*Continued on next page*

*Continued*

| Reagent type (species) or resource | Designation | Source or reference | Identifiers | Additional information |
|---|---|---|---|---|
| Cell line (*Drosophila*) | S2 GFP-Aurora B/WT-Cyclin B1-mCherry | This work | | |
| Cell line (Human) | HeLa Cyclin B1-venus | Jonathon Pines | | |
| Cell line (Human) | U2OS | *Afonso et al., 2014* | | |
| Cell line (Human) | hTERT-RPE1 Cyclin B1-venus | Jonathon Pines | | |
| Plasmid for mRNA synthesis (mouse) | Cyclin B1-mCherry | *Pasternak et al., 2016* | | |
| Transfected construct (*Drosophila*) | pMT-GFP-Aurora B | *Afonso et al., 2014* | | |
| Transfected construct (*Drosophila*) | non-degradable Cyclin B1-GFP | *Afonso et al., 2014* | | |
| Transfected construct (*Drosophila*) | WT Cyclin B1-GFP | *Mathieu et al., 2013* | | |
| Transfected construct (*Drosophila*) | KEN Cyclin B1-GFP | This work | | |
| Antibody | rabbit polyclonal anti *Drosophila* Cyclin B1 | James Wakefield | | 1:2500 |
| Antibody | mouse monoclonal anti Human Aurora B | Aim1, BD Biosciences | RRID:AB_2227708 | 1:500 |
| Antibody | rabbit polyclonal anti *Drosophila* Cenp C | Claudio Sunkel | | 1:10000 |
| Antibody | mouse monoclonal anti-α-tubulin (B-512 clone) | Sigma | | 1:2000 |
| Antibody | Alexa-Fluor secondary antibodies | Thermo Fisher | | 1:2000 |
| Commercial assay or kit | site-directed mutagenesis kit | Agilent | | |
| Chemical compound, drug | RO3306 | Sigma | | 10 µM |
| Chemical compound, drug | Binucleine-2 | Sigma | | 40 µM |
| Chemical compound, drug | ZM447439 | Tocris Bioscience | | 4–5 µM |
| Chemical compound, drug | MG132 | Merck | | 20 µM |
| Chemical compound, drug | Apcin | Tocris Bioscience | | 200 µM |
| Chemical compound, drug | pro-TAME | Boston Biochem | | 6.3 µM |
| Chemical compound, drug | SiR-DNA | Spirochrome | | 50 nM (Human cells) and 80 nM (*Drosophila*) |
| Chemical compound, drug | Sir-TUB | Spirochrome | | 20 nM |
| Software, algorithm | Prism V8 | Graphpad | RRID:SCR_002798 | |
| Software, algorithm | Fiji | ImageJ/Fiji | RRID:SCR_002285 | |
| Software, algorithm | Matlab | The MathWorks | RRID:SCR_001622 | |

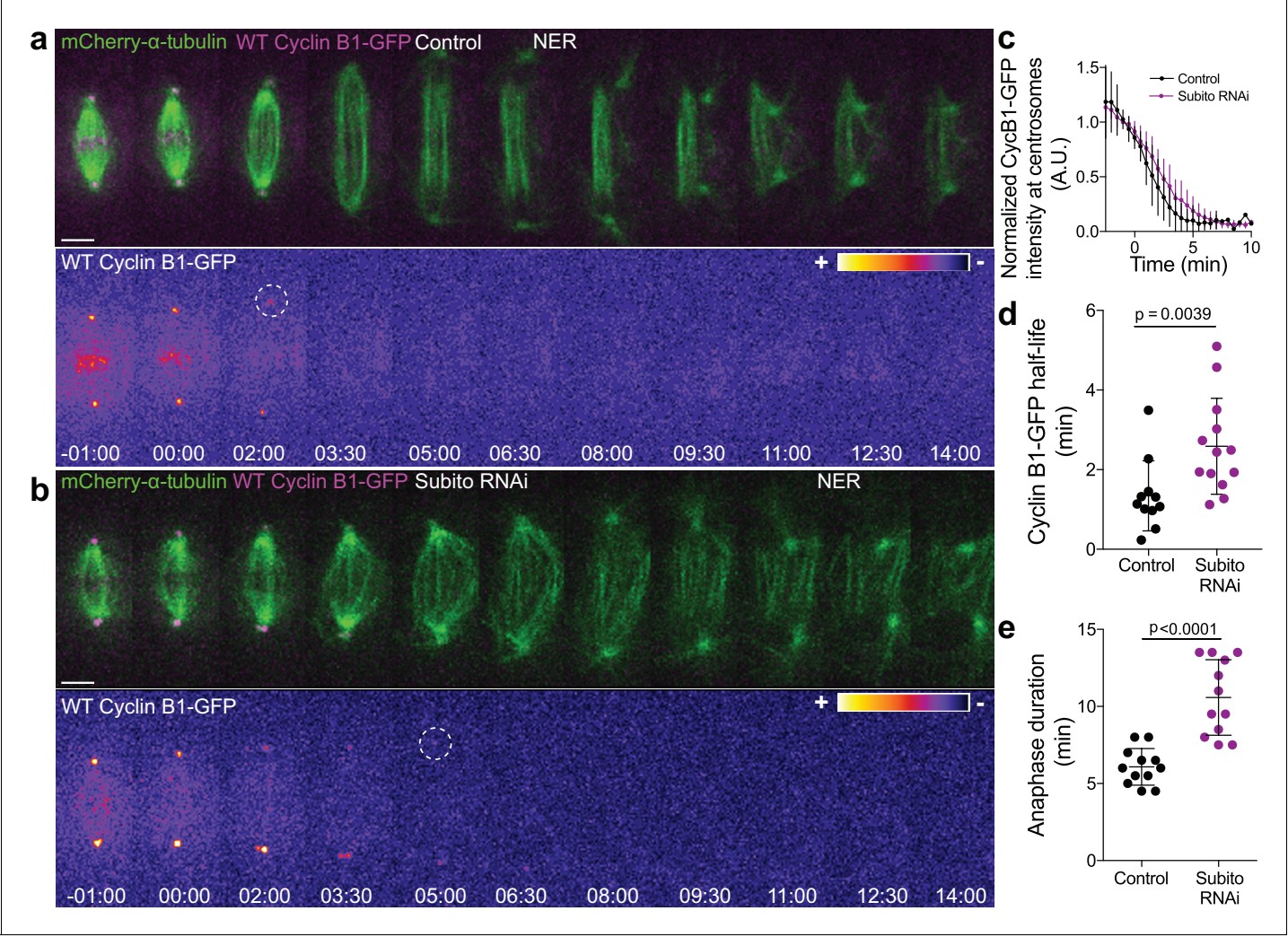

**Figure 8.** Preventing Aurora B localization at the spindle midzone delays Cyclin B1 degradation during anaphase. (**a**) and (**b**) Control and Subito/Mklp2-depleted *Drosophila* S2 cells stably expressing Cyclin B1-GFP/mCherry-α-tubulin. Cyclin B1-GFP localization is highlighted with the LUT 'fire' and dashed white circles highlight the frame before Cyclin B1 signal disappearance from centrosomes. Scale bars are 5 μm. Time is in min:sec. (**c**) Degradation profile of Cyclin B1-GFP quantified by measuring fluorescence intensity at centrosomes in control (n = 11 cells) and Subito/Mklp2-depleted S2 cells (n = 13 cells, pooled from three independent experiments). (**d**) and (**e**) Calculated Cyclin B1-GFP half-life (0–4.5 min after anaphase onset) and quantified anaphase duration, respectively, in the same control and Subito/Mklp2-depleted S2 cells as in (**c**). Statistical significance was tested with a nonparametric Mann–Whitney test and an unpaired t-test for data in (**d**) and (**e**), respectively.

DOI: https://doi.org/10.7554/eLife.47646.034

The following source data is available for figure 8:

**Source data 1.** Anaphase duration after Subito RNAi.

DOI: https://doi.org/10.7554/eLife.47646.035

## Cell culture

*Drosophila* S2 cells were cultured in Schneider medium supplemented with 10% FBS and grown at 25˚C. For live imaging cells were plated 2–3 hr before imaging in MatTek dishes (MatTek Corporation) pre-coated with 0.25 mg/ml concanavalin A. Human U2OS and HeLa cells were cultured in DMEM supplemented with 10% FBS and grown in a 5% $CO_2$ atmosphere at 37˚C. hTERT-RPE1 cells were grown in DMEM/F12 medium supplemented with 10% FBS. For all human cell lines, culture medium was changed to L-15 supplemented with 10% FBS 4–5 hr before live imaging. All cell lines used were free from mycoplasma contaminations as inferred by routine tests in a certified laboratory.

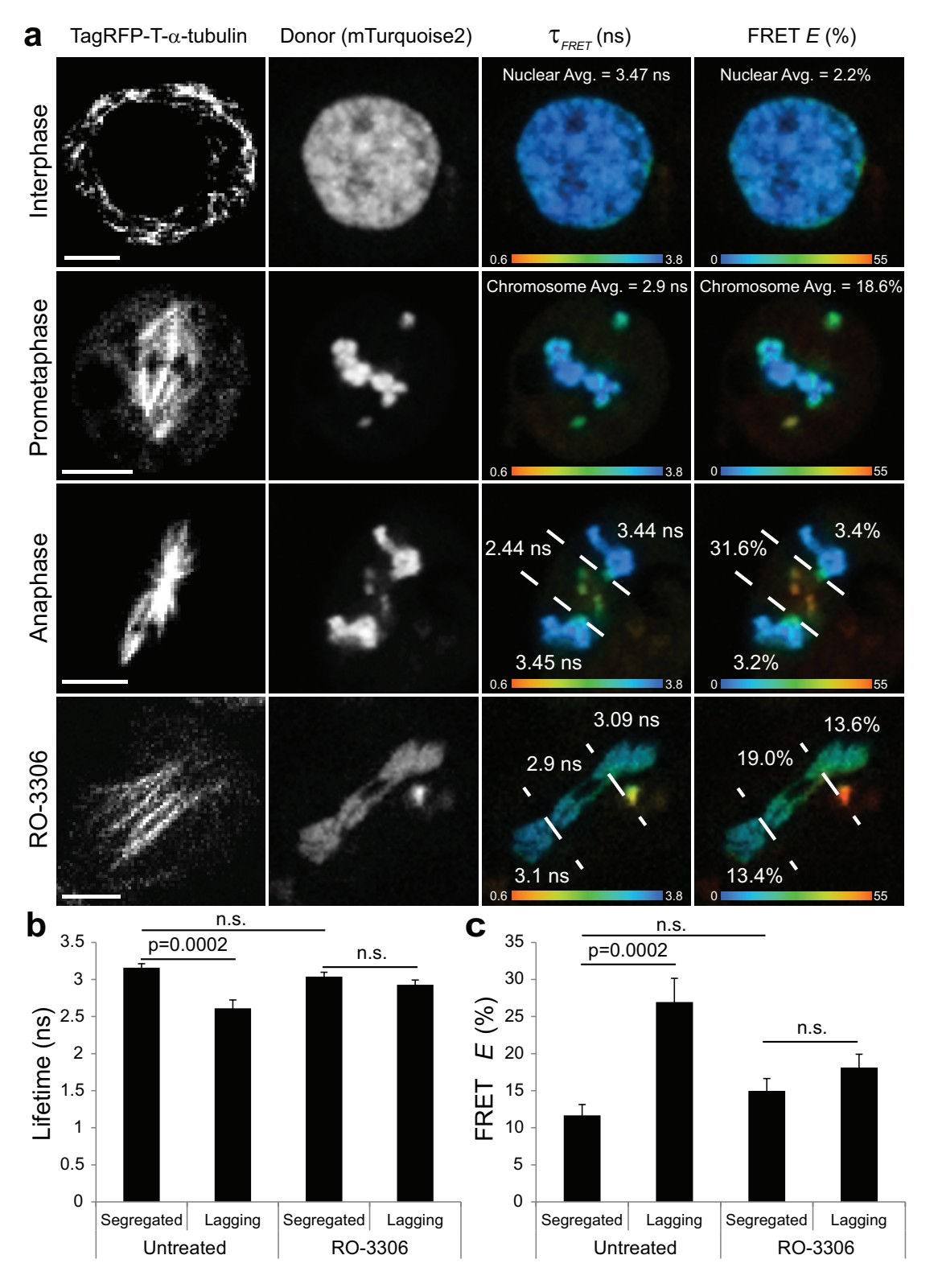

**Figure 9.** Visualization of a CyclinB1-Cdk1 activity gradient in the vicinity of the midzone during anaphase. (**a**) The chromatin targeted reporter exhibits a longer lifetime and lower FRET efficiency in interphase versus prometaphase. The sensor on lagging chromosomes in the vicinity of the midzone MTs, demarcated by the dashed lines, exhibits higher FRET efficiency and shorter donor lifetime than the reporter on the segregated DNA. The activity gradient is no longer evident in the presence of 10 μM RO-3306. (**b**) Quantification of donor lifetimes for segregated versus lagging chromatin in

*Figure 9 continued on next page*

*Figure 9 continued*

untreated and RO-3306-treated cells (N = 14 cells for each condition). (**c**) Quantification of FRET efficiencies for segregated versus lagging chromatin in untreated and RO-3306-treated cells (N = 14 cells for each condition). Error bars are SEM. Scale bars are 5 μm. Color wedges indicate 0.6 ns (blue) – 3.8 ns (red) for donor lifetimes and 0% (blue) to 55% (red) for FRET *E*. Two-tailed P-values from a Student's t-test are reported. n.s. is not significant (p>0.05).

DOI: https://doi.org/10.7554/eLife.47646.036

The following source data is available for figure 9:

**Source data 1.** Mean lifetimes, percentage of fluorophores exhibiting short and long lifetimes, and the short lifetimes for the chromatin-targeted Cyclin B1-Cdk1 activity reporter.
DOI: https://doi.org/10.7554/eLife.47646.037

**Source data 2.** Mean FRET efficiency statistics of chromatin-targeted Cyclin B1-Cdk1 FRET sensors.
DOI: https://doi.org/10.7554/eLife.47646.038

## Drug treatments

For acute inhibition of Cdk1 in S2 and U2OS cells the Cdk1 inhibitor RO3306 (Sigma) was added in metaphase or at anaphase onset, depending on the experimental set-up, at final concentration of 10 μM. Binucleine-2 (Sigma) was used to acutely inhibit Aurora B activity in *Drosophila* S2 cells at a final concentration of 40 μM in all experiments. BI2536 (Axon MedChem) was used in *Drosophila* cells at a final concentration of 100 nM. To reduce anaphase poleward chromosome segregation velocity, *Drosophila* S2 cells were incubated with 10 nM taxol (Sigma) prior to live imaging. For the live MG132 (Calbiochem) assay in *Drosophila* S2 cells, metaphase cells grown in the absence of the drug were selected in a multipoint set-up and MG132 was added after the first round of acquisition (5 min), at a final concentration of 20 μM. Cells that entered anaphase in a time window between 30 and 60 min after drug addition became arrested in anaphase, while the remaining cells became arrested in metaphase. MG132 addition at anaphase onset had no effect on the anaphase-telophase transition, likely due to a slower uptake of the drug in S2 cells. For the MG132 assay in U2OS and hTERT-RPE1 cells, MG132 was added 1–2 min before or at anaphase onset also at a final concentration of 20 μM. Drug addition 1–2 min after anaphase onset in U2OS or hTERT-RPE1 cells had no effect on anaphase duration. APC/C inhibition at anaphase onset on hTERT-RPE1 cells was achieved with a combination of both pro-TAME (Boston Biochem) and Apcin (Tocris Bioscience) used at final concentrations of 6.3 μM and 200 μM, respectively, as previously reported (*Sackton et al., 2014*). Similarly to MG132 treatment, APC/C inhibition 1–2 min after anaphase onset, or individual treatment with each drug showed a mild or no effect. APC/C inhibition in *Drosophila* cells was achieved by addition of Apcin at anaphase onset at a final concentration of 200 μM. ZM447439 (Tocris Bioscience) was used at 4 μM to inhibit Aurora B in U2OS cells and 5 μM for Aurora B inhibition at anaphase onset on hTERT-RPE1 cells. SiR-DNA (Spirochrome) was used at a final concentration of 80 nM in *Drosophila* S2 cells and 50 nM in hTERT-RPE1 cells and incubated 30–60 min prior to imaging. SiR-Tubulin (Spirochrome) was used at a final concentration of 20 nM and incubated 30–60 min prior to imaging.

## Immunofluorescence microscopy

*Drosophila* S2 cells were grown in coverslips previously pre-coated with 0.25 mg/ml concanavalin A. hTERT-RPE1 control and Mklp2-depleted cells were plated on glass coverslips. Cells were fixed with 4% PFA for 10 min, permeabilized with 0.3% Triton diluted in PBS for 10 min and washed 3 × 5 min with PBS-Tween. Primary antibodies used were mouse anti-α-tubulin (1:2000; B-512 clone, Sigma); rabbit anti-Cyclin B1 (gift from James Wakefield, University of Exeter); mouse anti-Aurora B (1:500, Aim1, BD Biosciences) and rabbit anti-Cenp C (gift from Claudio Sunkel, i3S, University of Porto, Portugal). Corresponding Alexa-Fluor secondary antibodies were used at 1:2000. Fixed images from *Figure 1—figure supplement 1* were obtained on an inverted Zeiss 780 confocal microscope with a 63x oil immersion objective. Fixed images from *Figure 1—figure supplement 3e and f* were obtained on an AxioImager Z1 with a 63x, plan oil differential interference contrast objective lens, 1.4 NA (from Carl Zeiss), equipped with a charge-coupled device (CCD) camera (ORCA-R2; Hamamatsu Photonics). Forced mitotic exit was achieved by a 10 min incubation with 10 μM RO3306 before fixation.

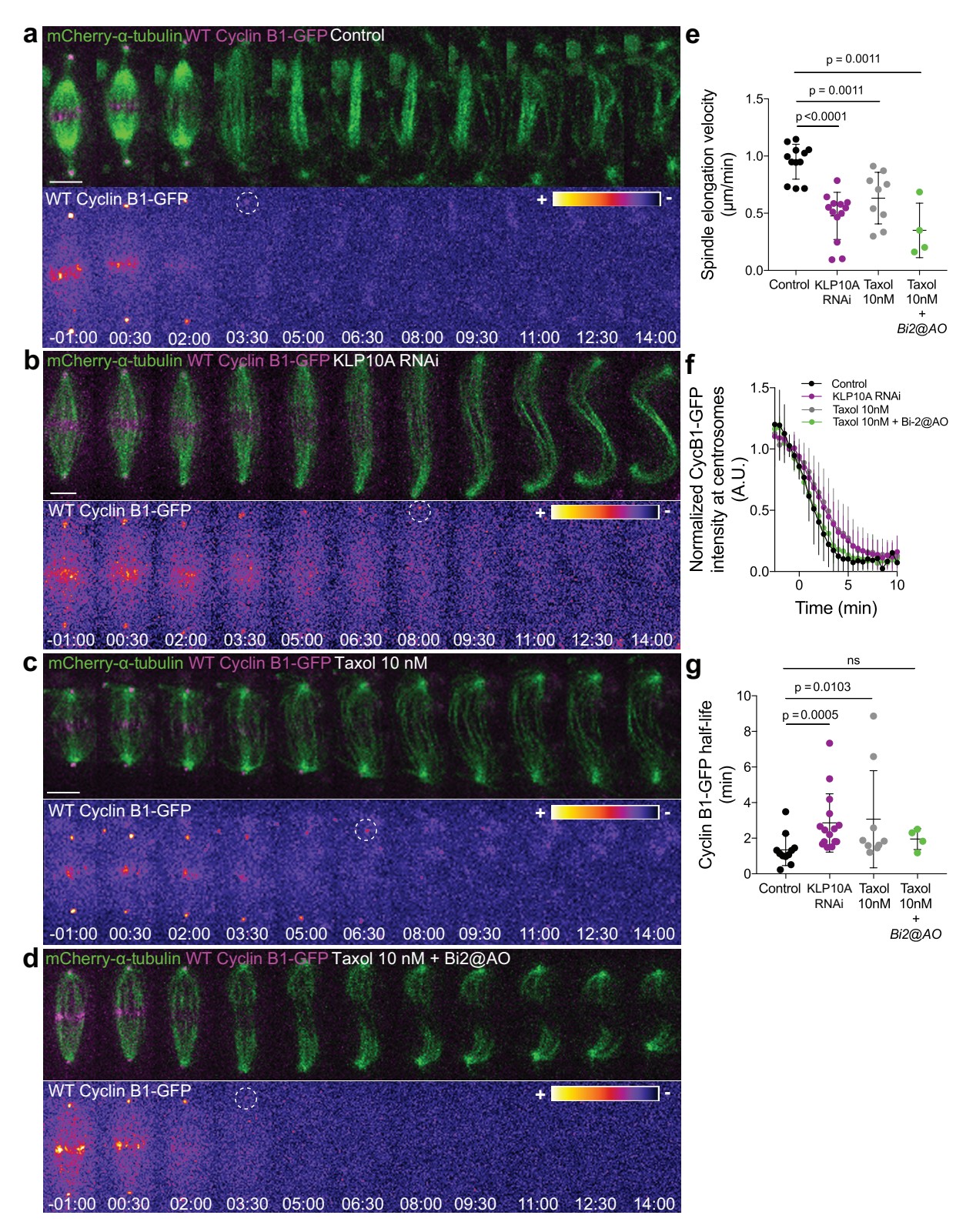

**Figure 10.** Cyclin B1 degradation during anaphase responds to slow chromosome separation in an Aurora B-dependent manner. (**a**) Control S2 cell stably expressing Cyclin B1-GFP/mCherry-α-tubulin. (**b**) KLP10A depleted S2 cell stably expressing Cyclin B1-GFP/mCherry-α-tubulin. (**c**) and (**d**) Examples of S2 cells treated with 10 nM Taxol during metaphase, where Binucleine-2 was added at anaphase onset in the condition shown in (**d**). For all conditions Cyclin B1 localization is highlighted with LUT 'fire' and dashed white circles highlight the frame before Cyclin B1 signal disappearance. *Figure 10 continued on next page*

Figure 10 continued

Scale bar is 5 µm. Time is in min:sec. (e) Half-spindle elongation velocity in control (n = 11 cells), KLP10A (n = 13 cells, pooled from three independent experiments), 10 nM Taxol (n = 9 cells) and 10 nM Taxol + Binucleine-2 addition at anaphase onset (n = 4 cells). Note the strong impairment of anaphase chromosome separation in all conditions. (f) and (g) Cyclin B1-GFP degradation profile and calculated Cyclin B1-GFP half-life in control (n=11 cells), KLP10A (n=15 cells, pooled from 3 independent experiments), 10 nM Taxol (n=9 cells) and 10 nM Taxol + Binucleine-2 addition at anaphase onset (n=4 cells).

DOI: https://doi.org/10.7554/eLife.47646.039

The following source data and figure supplements are available for figure 10:

**Source data 1.** Cyclin B1-GFP half-life after attenuation of chromosome separation velocity.
DOI: https://doi.org/10.7554/eLife.47646.043

**Figure supplement 1.** Aurora B or phosphatase inhibition at anaphase onset does not affect Cyclin B1 degradation kinetics.
DOI: https://doi.org/10.7554/eLife.47646.040

**Figure supplement 2.** Short exposure to low doses of Taxol do not compromise Aurora B localization at the spindle midzone.
DOI: https://doi.org/10.7554/eLife.47646.041

**Figure supplement 2—source data 1.** Time of GFP-Aurora B localization at the midzone after Taxol treatment.
DOI: https://doi.org/10.7554/eLife.47646.042

## Constructs

The pMT-GFP-Aurora B and the non-degradable Cyclin B1-GFP were described previously (*Afonso et al., 2014*). The non-degradable Cyclin B3 construct was a gift from Christian F. Lehner (University of Zurich, Switzerland). The KEN-box mutant was generated in the pMT-Cyclin B1-GFP plasmid using a site-directed mutagenesis kit (Agilent), following the manufacturer's instructions. The following primers were used for mutagenesis: Fw: 5'-ggacattgatgccaatgacgcggcgaacctgg-tactggtctcc-3' and Rv 5'- ggagaccagtaccaggttcgccgcgtcattggcatcaatgtcc-3'. To make the chromatin targeted FLIM-FRET sensor, DNA encoding the polo box domain (PBD) of polo like kinase 1 (Plk1), a 15 amino acid linker domain, and 16 amino acid recognition sequence containing the CyclinB1-Cdk1 auto-phosphorylation sites from human CyclinB1 was amplified by PCR from a previously generated CyclinB1-Cdk1 FRET sensor (*Gavet and Pines, 2010*), Addgene plasmid #2327) with a 5' SpeI site and 3' NotI site. The resulting PCR product was inserted by Gibson cloning between mTurquoise2 (FRET donor) and mVenus (FRET acceptor) in a pMT/V5 His-B vector backbone (Invitrogen) containing the CENP-C promotor inserted between the first XbaI site and the KpnI site and the human histone H2B gene inserted at the KpnI site downstream of the CENP-C promoter. The mTurquoise2 gene was positioned after the human histone H2B gene between KpnI and SpeI, while the mVenus gene with a stop codon was inserted between the NotI and SacII sites. To generate the non-phosphorylatable chromatin-targeted FRET sensor, the same construct was generated but using Addgene plasmid #2328 (*Gavet and Pines, 2010*) in which the codons for the phospho-serine residues were replaced with codons for alanines. The resulting constructs are well-suited for FLIM analysis due to the fact that mTurquoise2 exhibits a mono-exponential decay in its lifetime.

## Cell lines, transfections, and RNAi

The H2B-GFP/mCherry-α-tubulin and Lamin B-GFP/mCherry-α-tubulin stable cell lines have been previously described (*Afonso et al., 2014*). The WT-Cyclin B1-GFP/mCherry-α-tubulin, KEN-Cyclin B1-GFP/mCherry-α-tubulin and GFP-Aurora B/WT-Cyclin B1-mCherry stable cell lines were established using Effectene (Qiagen) according to the manufacturer's instructions. For visualization of WT-Cyclin B1-GFP, WT-Cyclin B1–mCherry and GFP-Aurora B signals 250 µM of $CuSO_4$ was added to the medium overnight (8–12 hr) to induce expression from a metallothionein promoter. With this copper concentration, we obtained a mild expression of the constructs without any detectable impact on mitotic progression. Aurora B overexpression was achieved by adding 1 mM of $CuSO_4$ instead of 250 µM for the same incubation period. The chromatin-targeted CyclinB1-Cdk1 FRET reporter was co-transfected with TagRFP-T-α-tubulin and selected with Blasticidin and then Hygromycin (Life Technologies). Some FLIM data were acquired from transiently transfected cells prior to selection and from stable cell lines that had been incubated overnight with 500 µM CuSO4, which still induced moderate expression even with the CENP-C promoter in place. The HeLa and hTERT-RPE1 Cyclin B1-Venus cell lines (*Collin et al., 2013*) were a kind gift from Jonathon Pines (Institute of Cancer Research, UK). The H2B-mRFP expression construct for human cells was obtained from

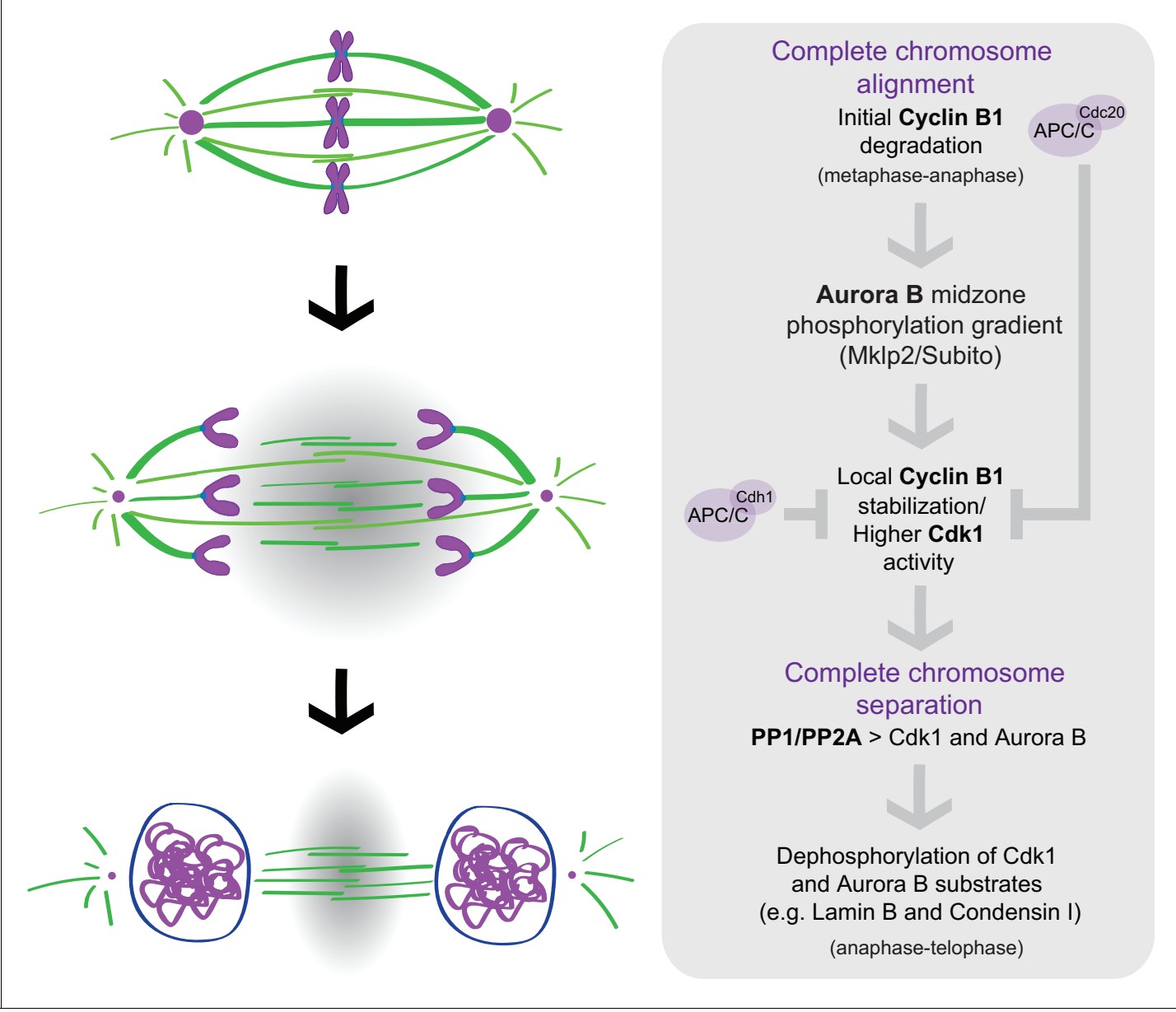

**Figure 11.** A crosstalk between molecular 'rulers' (Aurora B) and 'clocks' (Cdk1) licenses mitotic exit only after proper chromosome separation. APC/C$^{Cdc20}$ mediates the initial degradation of Cyclin B1 as chromosome align at the spindle equator and cells enter anaphase under SAC control. The consequent decrease in Cdk1 activity as cells enter anaphase targets Aurora B to the spindle midzone (via Subito/Mklp2/kinesin-6); Aurora B at the spindle midzone (counteracted by PP1/PP2A phosphatases on chromatin) establishes a phosphorylation gradient that locally delays APC/C$^{Cdc20}$- and APC/C$^{Cdh1}$-mediated degradation of residual Cyclin B1, and possibly Cdk1, at the spindle midzone. Consequently, as chromosomes separate and move away from the spindle midzone, Cdk1 activity decreases, allowing the PP1/PP2A-mediated dephosphorylation of Cdk1 and Aurora B substrates (e.g. Lamin B and Condensin I) necessary for mitotic exit.
DOI: https://doi.org/10.7554/eLife.47646.044

Addgene (#26001), transfected to 293 T cells to produce viruses and infect HeLa Cyclin B1-Venus expressing cells to generate a stable cell line. The human Cdk1-GFP construct was obtained from Origene (#RG200495) and transiently transfected into HeLa cells using Lipofectamin 2000 (Invitrogen) following the manufacturer's instructions. The H2B-GFP/mCherry-α-tubulin U2OS cell line, the KLP10A and the Subito/Mklp2 RNAi in *Drosophila* S2 cells were previously described (*Afonso et al., 2014*). Fzr RNAi in *Drosophila* S2 cells was achieved after two rounds of RNAi treatment for a total of 7 days. The following primers were used for dsRNA synthesis: 5'-

taatacgactcactatagggcaccggataatcaatacttggc-3'  and  5'-taatacgactcactatagggattcagaacg-gacttgttctcc-3'. Mklp2 depletion in hTERT-RPE1 cells was achieved 72 hr after transfection of siRNA oligonucleotides using Lipofectamine RNAiMAX (Invitrogen) following the manufacturer's instructions. The oligonucleotide sequence used was: 5'- AACGAACUGCUUUAUGACCUA-3'.

### Time-lapse microscopy

Live imaging data in *Figure 2* and *Figure 2—figure supplement 1* were obtained from a spinning disc confocal system (Andor Technology, South Windsor, CT) equipped with an electron multiplying CCD iXonEM+ camera and a Yokogawa CSU-22 U (Yokogawa Electric, Tokyo) unit based on an Olympus IX81 inverted microscope (Melville, NY). Two laser lines (488 and 561 nm) were used for near simultaneous excitation of GFP and mCherry/mRFP and the system was driven by Andor IQ software. Time-lapse image stacks of 0.8 μm steps were collected every 30 s, with a Plan- APO 100x/1.4 NA oil objective, with the exception of the experiments in *Figure 2—figure supplement 1* where image stacks were obtained every 60 s. Live imaging data in all other figures, except the FLIM experiments, were acquired in a temperature-controlled Nikon TE2000 or a Ti microscope, both equipped with a Yokogawa CSU-X1 spinning-disc head, imaged on an Andor iXon+ DU-897E EM-CCD. Excitation comprises three lasers (488 nm, 561 nm and 647 nm) that were shuttered by an acousto-optic tunable filter (TE2000) or electronically (Ti). Sample position was controlled via a SCAN-IM Marzhauser stage and a Physik Instrumente 541.ZSL piezo (TE2000) or via a Prior Scientific ProScan stage (Ti). Imaging of *Drosophila* S2 cells was performed with a 100x Plan-Apo DIC CFI Nikon objective in all experiments. Imaging of human cells and the *Drosophila* follicular epithelium was done with an oil-immersion 60 × 1.4 NA Plan-Apo DIC CFI (Nikon, VC series). Imaging of the mouse oocytes was performed either with a 60 × 1.27 NA CFI Plan-Apo IR water-immersion objective (Nikon) or a 40 × 1.30 NA Plan-Fluor oil-immersion objective (Nikon). Images were acquired with a 1 μm z-stack and 30 s time-lapse interval for all live imaging experiments in *Drosophila* S2 cells, *Drosophila* follicular epithelium and human cells. Mouse oocytes were acquired with a 4 μm z-stack and a 2 min time interval. Data from *Figures 3* and *6* were obtained with a 5 min time interval and data from *Figure 4* were obtained with a 1 min time interval. A temperature-controlled chamber was set up to 25°C for *Drosophila* live imaging or to 37°C for live imaging of human cells and mouse oocytes.

### Live-cell FLIM-FRET

Cells were imaged on a Nikon TiE A1R Spectral Detector Confocal with time-correlated FLIM at the UMass Amherst Institute for Applied Life Science Light Microscopy Facility using the 1.45 NA 100x oil immersion objective lens (Nikon). Time-correlated single photon counting (TCSPC) was conducted using two HPM-100–40 detectors (Becker and Hickl) and a 50 mHz pulsed 445 nm laser (BDL-445-SMN). Cells were identified by eye under wide-field fluorescence on the RFP channel. To collect FLIM data, the 445 nm pulsed diode laser (Becker and Hickl) was set to 50% laser power, and 40–50 million photons were detected per scan. NIS-Elements (Nikon) was used to control the microscope hardware, SPCM64 (Becker and Hickl) was used for FLIM acquisition, and SPCImage (Becker and Hickl) was used for FLIM analysis.

### FLIM analysis

FLIM analysis was conducted in SPCImage software to determine values for the fluorescence lifetime ($\tau_{\text{FRET}}$), the short lifetime, % short lifetime, % long lifetime, and FRET efficiency (E). FRET efficiency can be calculated using the equation:

$$E = 1 - \frac{\tau_{\text{FRET}}}{\tau_{\text{D}}} \times 100\%$$

Where $\tau_{\text{D}}$ is the lifetime of the donor under non FRET conditions, which was measured to be 3573.2 picoseconds in *Drosophila* S2 cells expressing mTurquoise2 in the absence of any acceptor under the same conditions used in the experiments. This value was fixed for the long lifetime when conducting 2-exponential fits in SPCImage to yield $\tau_{\text{FRET}}$ - the lifetime of the donor under possible FRET conditions. Fitting the data with a 2-exponential decay also provides values for the short lifetime, % short lifetime (indicative of the % of sensor engaged in FRET), and % long lifetime (indicative

of the % of sensor not engage in FRET). Decay matrices were generated to display color images of either fluorescence lifetimes or FRET efficiencies. For quantifying lagging and segregated chromatin, the donor lifetime and FRET efficiencies were quantified by drawing an ROI around chromatin in the vicinity of the midzone for lagging chromatin and compared to the average of the lifetime and FRET efficiencies of the two masses of segregated chromosomes. To assess the contribution of Cdk1, cells were imaged between 5-40 min following the addition of 10 μM RO-3306 (Santa Cruz Biotechnology).

## *Drosophila* strains and egg chamber preparation

Flies homozygous for the eGFP protein trap insertion into *CycB1* loci (CC01846 [*Buszczak et al., 2007*]) and expressing His2Av-mRFP (genotype: *w*; *P{PTT-GC}CycB*$^{CC01846}$; *P{His2Av-mRFP}*/+) were generated to image Cyclin B1 expression at endogenous levels while monitoring mitotic progression. *Drosophila* egg chambers were dissected and mounted in the gas-permeable oil 10S VOL-TALEF (VWR chemicals) for live imaging.

## In vitro culture and micro-injection of mouse oocytes

All mice were maintained in a specific pathogen-free environment according to the Portuguese animal welfare authority (Direcção Geral de Alimentação e Veterinária) regulations and the guidelines of the Instituto de Investigação e Inovação em Saúde animal facility. Oocytes were isolated from ovaries of 7–10 week old CD-1 mice, cultured in M2 medium under mineral oil and matured to metaphase II in vitro for 16 hr at 37˚C. Oocytes were then microinjected with mRNA encoding Cyclin B1-mCherry (*Pasternak et al., 2016*) using an electric-assisted microinjection system (*FitzHarris et al., 2018*). A function generator (GW Instek AFG-2005) was used to produce electrical current instead of an intracellular electrometer. Oocytes were then transferred to M2 medium supplemented with SiR-DNA (500 nM, Spirochrome) and allowed to express the mRNA for 4–6 hr. Oocytes were then induced parthenogenically by rinsing into calcium-free M2 medium with 10 mM strontium chloride, and imaged. mRNA in vitro transcription pGEMHE-CyclinB1-mCherry (*Pasternak et al., 2016*) was a kind gift from Melina Schuh (Max-Planck Institute for Biophysical Chemistry). The plasmid was linearised, and capped mRNA was synthesized using the T7 ARCA mRNA kit (New England Biolabs) according to the user's manual and resuspended in water. The final concentration of mRNA in the injection needle was 400 ng/μl.

## Definition of mitotic exit

Mitotic exit was defined either by the moment of NER or chromosome decondensation. When a nuclear envelope marker was present (ex: Lamin B in *Figure 2*) mitotic exit was defined as the first frame of appearance of Lamin B in the nascent nucleus. In experiments where cells expressed mCherry-α-tubulin, NER was determined based on the first frame where soluble mCherry-α-tubulin was excluded from the main nucleus. In those cases where mCherry-α-tubulin was not present, SiR-DNA was used instead and visual inspection of DNA decondensation was used to define mitotic exit.

## Quantification of Cyclin B1 decay during anaphase

Image processing and quantification was performed using Fiji. Data from *Drosophila* S2 in *Figures 3e* and *5d* and h, 8 c, 10 f and *Figure 1—figure supplement 2d*, Cyclin B1 were measured with a constant circular ROI around the centrosomes. This resulted in more robust measurements, which were less prone to fluorescence fluctuations in the cytoplasm. This was essential to detect differences in the Cyclin B1 decay in the different conditions tested. Exceptionally, and to compare Cyclin B1 levels in *Drosophila* and human cells, in *Figure 1c* total Cyclin B1-GFP fluorescence intensity was measured using an ROI defined by the limits of the cell. Cyclin B1 levels from *Figure 7e,g* and *Figure 7—figure supplement 1* were measured with a constant rectangular ROI defined by the size of the metaphase plate before cells entered anaphase. This size and position was maintained to measure the intensity of Cyclin B1 that localized at the spindle midzone in control cells and after Cdk1 addition. Note that with the same ROI in *Figure 7—figure supplement 1c* both Cyclin B1 and Aurora B intensities were measured. Data from human cells from *Figures 1c* and *6f* and *Figure 1— figure supplement 3*, total Cyclin B1 were measured with an ROI defined by the limits of the cell. In

both cases, the ROI was used to manually track the centrosome/midzone/cell over time. Images were maximum intensity projected and background was measured for all time points with the same ROI used for Cyclin B1 measurements in a region outside the cell. The background levels were subtracted from the absolute intensity values. In the *Drosophila* follicular epithelium Cyclin B1 levels were measured in sum-intensity Z projections, using a circular ROI that was manually tracked in an homogeneous cytoplasmic region (free of accumulated Cyclin B1 at microtubules or centrosomes) of dividing cells. Background was measured with the equivalent ROI in G1 phase cells (identified by undetectable Cyclin B1-GFP expression) within the same egg chamber, and subtracted to dividing cells to establish the baseline background value. In mouse oocytes, Cyclin B1 was measured using a ROI placed manually into the center of the oocyte, background levels were measured in a ROI outside the cell and used for subtraction from the mean cytoplasmic intensity. Images were sum-intensity Z projected.

## Calculation of Cyclin B1 half-life

The half-life of Cyclin B1-GFP or Cyclin B1-Venus was calculated based on the formula

$$t * \frac{ln2}{\ln\left(\frac{n(t)}{n(0)}\right)}$$

where $t$ is the time interval, $n(0)$ is the background subtracted Cyclin B1-GFP fluorescent intensity at anaphase onset and n(t) is the background subtracted Cyclin B1-GFP fluorescent intensity at time 4.5 min or 6.5 min after anaphase onset, as indicated in the figure legends.

## Statistical analysis

Normality of the samples was determined with a D'Agostino and Pearson test. Statistical analysis for two-sample comparison, with normal or non-normal distribution, was performed with a t-test or Mann-Whitney test, respectively. P value was considered extremely significant if $p<0.0001$ or $p<0.001$, respectively, very significant if $0.001 < P < 0.01$ and significant if $0.01 < P < 0.05$. In all plots error bars represent standard deviation, unless indicated otherwise. All statistical analysis was performed with GraphPad Prism V8 (GraphPad Software, Inc), except the calculation of P-values in the FLIM experiments, which used the PlotsOfDifferences web app (*Goedhart, 2019*).

## Sample size and replicates

The sample size was not statistically determined. Whenever possible a minimum of n = 10 was analyzed, however, in cases where the data obtained were highly consistent an n < 10 was considered. For RNAi-mediated depletion experiments at least two independent RNAi experiments were prepared and imaged. Statistical differences between independent RNAi experiments were analyzed, and whenever sample size was not large enough cells from independent experiments were pulled all together and statistical analysis was performed on the entire population. For drug addition experiments each cell was considered an independent experiment and a minimum of 4 or more cells were imaged for each condition.

## Code availability

Matlab based custom routines were used for the generation of kymographs. The Source Code File is provided as supplementary information.

## Data availability

All data generated or analyzed during this study are included in this published article (and its supplementary information files).

## Acknowledgements

We thank Eric Griffis, Jean-René Huynh, Claudio Sunkel, Jonathon Pines, Melina Schuh and Christian Lehner for the kind gift of reagents, and Marco Gonzalez-Gaitán for supporting OA during the final stages of this work. LPC is the recipient of a Marie Skłodowska-Curie Action fellowship (grant agreement 746515). EMS holds an FCT Investigator position and his work is supported by Fundação para

a Ciência e a Tecnologia (PTDC/BEX-BCM/0432/2014). This work was supported by R01GM107026 grant to TJM and a Commonwealth Honors College grant to CMC Confocal and FLIM microscopy data collection was performed in the Light Microscopy Facility and Nikon Center of Excellence at the Institute for Applied Life Sciences, University of Massachusetts Amherst with support from the Massachusetts Life Science Center. Work in the HM lab is supported by the European Research Council (ERC) under the European Union's Horizon 2020 research and innovation programme (grant agreement No 681443) and FLAD Life Science 2020.

# Additional information

### Funding

| Funder | Grant reference number | Author |
|---|---|---|
| European Research Council | 681443 | Helder Maiato |
| Fundação Luso-Americana para o Desenvolvimento | FLAD LifeScience2020 | Helder Maiato |
| H2020 Marie Skłodowska-Curie Actions | 746515 | Liam Cheeseman |
| Fundação para a Ciência e a Tecnologia | PTDC/BEX-BCM/0432/2014 | Eurico Morais-de-Sa |

The funders had no role in study design, data collection and interpretation, or the decision to submit the work for publication.

### Author contributions

Olga Afonso, Formal analysis, Validation, Investigation, Visualization, Writing—original draft, Writing—review and editing; Colleen M Castellani, Formal analysis, Validation, Investigation, Visualization; Liam P Cheeseman, Eurico Morais-de-Sá, Formal analysis, Investigation, Writing—review and editing; Jorge G Ferreira, Bernardo Orr, Formal analysis, Investigation, Visualization; Luisa T Ferreira, Resources; James J Chambers, Resources, Visualization, Methodology; Thomas J Maresca, Conceptualization, Resources, Data curation, Formal analysis, Supervision, Funding acquisition, Project administration, Writing—review and editing; Helder Maiato, Conceptualization, Resources, Data curation, Supervision, Funding acquisition, Writing—original draft, Writing—review and editing, Project administration

### Author ORCIDs

Thomas J Maresca http://orcid.org/0000-0003-2214-8674
Helder Maiato https://orcid.org/0000-0002-6200-9997

### Ethics

Animal experimentation: All mice were maintained in a specific pathogen-free environment according to the Portuguese animal welfare authority regulations (Direcção Geral de Alimentação e Veterinária; reference# 0421/000/000/2016) and the guidelines of the Instituto de Investigação e Inovação em Saúde animal facility.

### Decision letter and Author response

Decision letter https://doi.org/10.7554/eLife.47646.051
Author response https://doi.org/10.7554/eLife.47646.052

# Additional files

### Supplementary files

• Source code 1. Kymograph generation.
DOI: https://doi.org/10.7554/eLife.47646.045

- Supplementary file 1. Conservation of D-box, KEN boxes and Aurora B phosphorylation sites on Drosophila Cyclin B1 and human Cyclins B1 and B2.
DOI: https://doi.org/10.7554/eLife.47646.046

- Transparent reporting form
DOI: https://doi.org/10.7554/eLife.47646.047

### Data availability

All data generated or analysed during this study are included in the manuscript and supporting files.

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
