## [Decision Letter]

Thank you for submitting your article "Mitotic exit is controlled during anaphase by an Aurora B-Cyclin B1/Cdk1 crosstalk" for consideration by *eLife*. Your article has been reviewed by Anna Akhmanova as the Senior Editor, a Reviewing Editor, and two reviewers. The reviewers have opted to remain anonymous.

The reviewers have discussed the reviews with one another and the Reviewing Editor has drafted this decision to help you prepare a revised submission.

Summary:

Afonso et al., present evidence for a mitotic exit network in *Drosophila* S2 cells regulated by crosstalk between Aurora B and Cyclin B-Cdk1 in anaphase. They show that low levels of cyclin B persist in early anaphase and that cyclin B continues to be degraded during anaphase, most likely by APC/C-Cdh1. The timing of chromosome decondensation and nuclear envelope reformation correlate with the disappearance of cyclin B1 from centrosomes and preventing cyclin B degradation in anaphase using the proteasome inhibitor MG132 arrests cells in early anaphase and delays NER and chromosome decondensation. This arrest depends on Cdk1 activity. They further show that a small amount of cyclin B1-GFP is present on the spindle midzone in anaphase in S2 cells, colocalizing with Aurora B and depending on Subito/Mklp2. In *Drosophila*, cyclin B1 is a substrate of Aurora B (Mathieu et al., 2013), and Afonso et al. show that the degradation of a phosphomimetic (5E) mutant of cyclin B1 is delayed, and increases anaphase duration. A non-phosphorylatable (5A) cyclin B mutant is degraded faster in anaphase. The authors propose that the initial degradation of cyclin B1 in metaphase drives Aurora B to the spindle midzone via Subito/Mklp2. Subsequently, Aurora B at the spindle midzone establishes a phosphorylation gradient that spatially restricts cyclin B1 localization and delays its degradation via APC/C cdh1.

This is an interesting model but the reviewers had several major concerns over the generality of the model – notably whether it was *Drosophila*-specific – and that some of the key experiments did not directly test the model that cyclin B1 degradation is responsive to physiological perturbation of mitotic exit conditions.

Essential revisions:

1) The authors should show that AurB inhibition delays Cyclin B degradation in *Drosophila* S2 cells and in endogenously tagged human cells.

2) The authors should show that endogenous Cyclin B localises to the midzone; moreover, there is no evidence provided that spindle midzone localization of cyclin B requires Aurora B (protein or activity). Subito depletion also affects localization of Polo at the midzone (Cesario et al., 2006), so cyclin B localization might as well depend on this kinase, or simply depend on Subito and not on Aurora B.

3) Since none of the Aurora B sites in *Drosophila* cyclin B1 are conserved in human cyclin B1 (Mathieu et al., 2013) the authors should analyse the effects of human AurB phospho-mutants.

Other points:

1) The way the Aurora B and Subito experiments are presented and interpreted are confusing. The model proposed in Figure 10 predicts that a lack of Aurora B at the spindle midzone would accelerate cyclin B degradation due to the absence of a phosphorylation gradient that spatially restricts cyclin B1 on the spindle midzone. However, in Figure 8, and Figure 7—figure supplement 1, cyclin B1 degradation is delayed when Aurora B is not on the midzone, and anaphase duration prolonged. This could be due to the fact that after Subito knock-down, Aurora B is retained much longer on the chromatin (Figure 1–figure supplement 3f), similar to what happens after Aurora B overexpression (Figure 7—figure supplement 1). Having Aurora B (and cyclin B/cdk1) at the midzone allows the formation of a phosphorylation gradient (due to counteracting activities of PP2A/PP1). Movement of chromosomes away from the heart of the gradient would allow exit. When Aurora B (and cyclin B? – this is unclear) is retained on the chromatin, this spatial separation is obviously not possible. Moreover, it might also explain why degradation of cyclin B at centrosomes is delayed under these conditions as more Aurora B is in closer proximity of these structures. Related to this: does cyclin B colocalize with Aurora B on the chromatin after Subito depletion? Where do the cyclin B 5E and 5A mutants localize (apart from the centrosomes)? Potential localization on chromatin is masked by the H2B-GFP.

2) It looks as if depletion of Mklp2 affects cyclin B localization on the mitotic spindle before anaphase. Is this consistent? If so, doesn't this impact the cyclin B-GFP intensity measurements?

3) The lack of effect of acute PP1/PP2A inhibition on cyclin B1 degradation (Figure 9G) is difficult to reconcile with a model in which a phosphorylation gradient concentrates cyclin B/cdk1 activity at the spindle midzone (Figure 10), and with the observation that a cyclin B 5E mutant is more stable and delaying mitotic exit (Figure 9C,D). The latter would suggest that cyclin B needs to be dephosphorylated for timely degradation.

4) All referees had concerns with Figure 4:

What is happening to the spindle and chromosomes after MG132 addition and subsequent Cdk1 inhibition in U2OS cells. It looks as if cytokinesis (abscission unclear) is taking place and that in these daughter cells the half spindle (00:10) is forming a new 'metaphase' spindle over time on which the separated sisters try to align (01:00 and 01:30). After Cdk1 inhibition, both spindles undergo an aberrant anaphase/exit. Can this phenotype be regarded as a delay in anaphase, or should it be considered as a reversion into prometaphase? In S2 cells, the anaphase delay is much more clear, because cytokinesis does not appear to take place and the two half spindles rejoin after Cdk1 inhibition.

If cells only enter anaphase when Cdk1 activity has fallen below the required threshold to maintain metaphase, then how can cells arrested in anaphase by MG132 return to a metaphase-like state? Presumably this state results from blocking degradation events (cyclin B or other targets) normally contributing to disassembly of the spindle poles during anaphase. (And it implies that the central spindle – in contrast – disassembles under these conditions). Without further justification of the 'metaphase' label (e.g. restoration of Cdk activity), it should be avoided.

Proteasomal inhibition in anaphase blocks mitotic exit. The authors interpret these results in an "anaphase arrest" This is true as far as cohesion cleavage is concerned. However, the spindle appears to reform and from a cytoskeletal and cell cycle control perspective it looks more as if the cells were simply being pushed back into a mitotic state. This experiment shows that if Cdk1 activity is maintained above a threshold mitotic exit is blocked, most likely because phosphatases fail to reactivate. The particular experimental set up (i.e. adding a proteasome inhibitor following anaphase onset) is novel; however, the results are not particularly surprising, given for example the results shown by Potapova et al. that even reactivation of Cdk1 in late telophase reactivates the mitotic state. The movies and images do not really support the notion of accelerated mitotic exit by AurB inhibition. Likewise, in the legend for Figure 4C the authors state: Note that for MG132 and MG132+Aurora B inhibition anaphase duration corresponds to the total duration of the movie as most cells do not exit mitosis until the end of the acquisition. I am not sure how in this case accelerated mitotic exit following AurB inhibition can be supported by the data? Also, it is not clear what the grey, green, purple colour code means.

5) The videos are not referred to in the text.

6) Degradation analyses of cyclin B1 usually show that cells enter anaphase with some protein remaining (in contrast to some figures in the original Clute & Pines 1999 study, which is cited to justify the statement at bottom of p8 that cyclin B1 is 'undetectable' at anaphase entry). The thresholding of anaphase entry underpins the phenomenon of mitotic slippage and has been explored at length in more recent literature that is not cited in this study. For example, a study from Stephen Taylor's lab (Gascoigne, 2009) measured a Cyclin B1-GFP threshold for anaphase corresponding to 32% of metaphase levels.

7) The phrase 'mitotic exit' needs to be used more carefully. It is conventionally used to describe the collective events that occur after anaphase onset and the authors sometimes use the phrase in this sense but in some places they use it to describe the point at which cells have completed these events (e.g. subsection “Cyclin B1 continues to be degraded during anaphase and its disappearance is a strong predictor of mitotic exit in metazoans” 'complete Cyclin B1 disappearance from centrosomes during anaphase strongly correlated with mitotic exit').

8) No mention is made of the likely presence of cyclin B3 in anaphase cells. B3 is degraded later than B1, and could contribute to Cdk1-dependent delay in mitotic exit.

9) Figure 5 shows a very small effect of FZR depletion on cyclin B1 degradation at anaphase (especially in (f)) suggesting that Cdh1 may only play a minor role. The authors argue that prolonged Cdc20 activity can compensate for lack of Cdh1 under these conditions, by targeting cyclin B1 through its D-box (whereas Cdh1 targets it through KEN). These arguments do not properly reflect the latest thinking on patterns of APC/C-degron recognition. Cryo-EM structures published over the recent years have established a paradigm of multivalent APC/C-substrate interaction through two or more degrons and Cdc20 and Cdh1 have almost identical KEN and D-box receptor sites. It would be safer to conclude that *Drosophila* Cyclin B1 degradation in anaphase is mediated through both KEN and D-box, and can be carried out by either Cdc20 or Cdh1. This could be tested by the combination of KEN-mt/APCin.

The authors should note that APCin cannot be expected to inhibit a KEN-box dependent event, since it is a D-box competitor, so the discussions at top of p6 and middle p10 are incorrect – the additive effect of APCin and proTAME on anaphase arrest does not support a role for Cdh1 in anaphase cyclin B1 degradation. It supports that additive interactions of Cdc20/Cdh1 and substrate degrons are required for efficient ubiquitination of substrates.

10) Effects on cyclin B1-GFP degradation look very small in regular degradation plots, and are more convincingly presented as half-life measurements (e.g. Figure 8E, Figure 1—figure supplement 3D). The degradation curve for OA treatment (Figure 9E) also appears to show a very small difference in rate between treated and untreated cells; these data should likewise be presented as half life measurements to provide same level of proof as in other figures.

11) As far as I am aware, 'anaphase arrest' (Figure 5J) has never been described as a consequence of FZR depletion in mammalian cells. Could additional markers be used to show that this is really an anaphase arrest (as opposed to death in anaphase)?

12) To my knowledge Cdk1 inhibition after anaphase onset has not been analysed before and it makes sense that this causes a faster reformation of the NE. This can be explained by the Cundell et al., papers due to premature Greatwall inactivation and PP2A:B55 reactivation. These papers should be cited here.

[Editors' note: further revisions were requested prior to acceptance, as described below.]

Thank you for resubmitting your work entitled "Spatiotemporal control of mitotic exit during anaphase by an Aurora B-Cdk1 crosstalk" for further consideration at *eLife*. Your revised article has been favorably evaluated by Anna Akhmanova (Senior Editor) and Jonathon Pines as Reviewing Editor.

The revised paper has addressed the substantive criticisms of the reviewers but before publication you will need to modify the text to take account of the following points, and in particular to tone down your conclusions for human cells:

1) Subsection “Cyclin B1 degradation during anaphase is mediated by APC/C^Cdh1^: Glotzer et al., 1991 is not the correct reference for Cyclin B1 degradation under the control of the SAC. The correct reference is Clute and Pines, 1999.

2) Subsection “Cyclin B1 degradation during anaphase is mediated by APC/C^Cdh1^: The caveat to the APCin/TAME experiment is that both inhibitors might be needed because they are relatively inefficient, not that Cdc20 and Cdh1 are additive in anaphase.

3) Subsection “Cyclin B1-Cdk1 localization and activity during anaphase are spatially 224 regulated by Aurora B at the spindle midzone”: The caveat to Figure 7 is that the presence of Cdk1 does not imply the presence of Cyclin B1.

4) Discussion section: APC/C-Cdc20 degrades Cyclin B1 in metaphase not at the meta-and transition.

5) Discussion section: You have not shown that human Cyclin B1 persists at the spindle midzone, just Cdk1.

---

## [Author Response]

This is an interesting model but the reviewers had several major concerns over the generality of the model – notably whether it was Drosophila-specific – and that some of the key experiments did not directly test the model that cyclin B1 degradation is responsive to physiological perturbation of mitotic exit conditions.Essential revisions:1) The authors should show that AurB inhibition delays Cyclin B degradation in Drosophila S2 cells and in endogenously tagged human cells.

We have performed the required experiments in both *Drosophila* S2 and endogenously tagged human (hTERT-RPE1) cells expressing fluorescent Cyclin B1. Accordingly, Aurora B was acutely inhibited at the first signs of sister chromatid separation at anaphase onset, either with Binucleine-2 (S2 cells) or with ZM447439 (hTERT-RPE1 cells). In both cases, Aurora B inhibition did not delay Cyclin B1 degradation during anaphase. These results are in agreement with our previous studies (Afonso et al., 2014), which showed that acute Aurora B inhibition at anaphase onset in both *Drosophila* and Human cells did not accelerate mitotic exit. These data suggest that Aurora B-mediated feedback mechanism is only required to delay Cyclin B1 degradation and consequently mitotic exit in the presence of incompletely separated chromosomes. To directly test this, first we delayed chromosome separation by treating S2 cells with low doses of taxol or by depleting KLP10A, as shown previously (Afonso et al., 2014). Both treatments induced a delay in Cyclin B1-GFP degradation. Remarkably, acute inhibition of Aurora B activity at anaphase onset in taxol treated cells restored normal Cyclin B1 degradation. These results are now included in the manuscript as new Figure 10. Taken together, these data suggest that Aurora B-mediated regulation of Cyclin B1 degradation is only required to extend anaphase duration in the presence of delayed chromosome segregation, consistent with a feedback mechanism that delays mitotic exit in the presence of incompletely separated chromosomes during anaphase.

2) The authors should show that endogenous Cyclin B localises to the midzone; moreover, there is no evidence provided that spindle midzone localization of cyclin B requires Aurora B (protein or activity). Subito depletion also affects localization of Polo at the midzone (Cesario et al., 2006), so cyclin B localization might as well depend on this kinase, or simply depend on Subito and not on Aurora B.

We have tried exhaustively to localize endogenous Cyclin B1 either by immunofluorescence or by endogenously tagging Cyclin B1 with a fluorescent reporter in several systems, but we were limited by technical difficulties due to the extremely low signal/noise ratio of Cyclin B1 during anaphase. We have inclusively modified the acquisition routine for live-cell recording of endogenously tagged Cyclin B1Venus in both human HeLa and hTERT-RPE1 cells in order to adjust exposure times to the gradually decreasing signal of Cyclin B1 as cells entered anaphase (see Author response image 1). While this unequivocally showed the presence and gradual decrease of Cyclin B1 during anaphase, the signal from mid-anaphase was so low that, with the new adjusted exposure times, the camera started picking auto-fluorescent cytoplasmic granules (confirmed in untagged parental hTERT-RPE1 cells, not shown here).

To overcome this technical difficulty, we started by forcing S2 cells to prematurely exit mitosis with higher endogenous levels of Cyclin B1 by acute Cdk1 inhibition in metaphase. Under these circumstances, signal/noise ratio improved significantly and we were able to co-localize endogenous Cyclin B1 with spindle midzone microtubules by immunofluorescence (see new Figure 1—figure supplement 1B,B’). Surprisingly, there is no published data that we are aware of regarding Cdk1 localization during mitosis in mammalian cells. As so, we have performed localization experiments of Cdk1-GFP at high spatial and temporal resolution in live human cells. We found that Cdk1-GFP decorated the spindle region at the metaphase-anaphase transition and was clearly enriched at the central spindle region as cells progressed throughout anaphase, accumulating at the midbody during cytokinesis. We also note that Cdk1 has recently been identified by mass-spectrometry on isolated midbodies from human cells (bioRxiv 569459; doi: https://doi.org/10.1101/569459). Therefore, although the low signal/noise ratio of endogenous Cyclin B1 did not allow us to unequivocally infer the localization of human Cyclin B1 during anaphase, we now show that human Cdk1 is enriched in the central spindle region during anaphase, and midbody during cytokinesis. These data are now provided in new Figure 7—figure supplement 3A,B).

In support of our model for spatial control of Cdk1 activity during anaphase, we have now optimized the conditions to monitor Cdk1 activity by adapting a previously reported Cdk1 FRET sensor (Gavet and Pines, 2010) for expression in S2 cells, followed by FLIM-FRET analysis. Strikingly, targeting the Cdk1 FRET sensor to chromatin by fusion with H2B histone revealed a gradient of Cdk1 activity on chromosomes, with lagging chromosomes showing significantly higher FRET than leading chromosomes during anaphase, as predicted by our model. This gradient was abolished by Cdk1 inhibition with RO-3306, and the respective non-phosphorylatable sensors did not report any significant Cdk1 activity. These data are now included in new Figure 9 and Figure 9 –source data 1 and Figure 9—source data 2. We concluded that Cdk1 activity on chromosomes is a function of chromosome separation during anaphase, with incompletely separated chromosomes positioned closer to the spindle midzone showing the highest Cdk1 activity during anaphase.

Finally, we used our metaphase Cdk1 inhibition essay to force entry into anaphase with higher levels of Cyclin B1 and monitored its localization at the spindle midzone after acute inhibition of Aurora B or Polo at anaphase onset in living S2 cells. Consistent with our previous Subito depletion experiments, these data showed that only Aurora B inhibition prevented Cyclin B1 to accumulate at the spindle midzone. Taken together, these data indicate that Aurora B localization and activity are required for Cyclin B1 accumulation at the spindle midzone in S2 cells.

3) Since none of the Aurora B sites in Drosophila cyclin B1 are conserved in human cyclin B1 (Mathieu et al., 2013) the authors should analyse the effects of human AurB phospho-mutants.

Given that there are several putative Aurora B phosphorylation sites on human Cyclin B1 and Cyclin B2 that are not conserved with *Drosophila* Cyclin B1 (and vice-versa; see below), this is not an easy enterprise to complete within just two months. We have nevertheless focused on two putative sites of human Cyclin B1, one (S310) that is conserved with human Cyclin B2 and *Drosophila* Cyclin B1, and another (S413) that is conserved between human Cyclin B1 and Cyclin B2 and has been previously identified as an Aurora B phosphorylation site on Cyclin B1 in large scale phospho-proteomic studies of mitosis (Kettenbach et al., 2011). These two sites were simultaneously mutated into Alanine (A) or Aspartic Acid (D) to respectively mimic non-phosphorylatable and constitutively phosphorylated human Cyclin B1 and transiently expressed as fusion proteins with GFP in human HeLa cells.

The CLUSTAL O(1.2.4) multiple sequence alignment can be found in Supplementary file 1.

Quantification of Cyclin B1-GFP fluorescence of both WT and respective mutants revealed only minor differences in the degradation kinetics of each version of Cyclin B1 during anaphase, but a clear effect of the phosphomimetic mutant already prior to anaphase (0 min).

**Author response image 2. respfig2:** 

While these experiments, together with existing literature (Kettenbach et al., Science Signal, 2011), do support a role for Aurora B in Cyclin B1 phosphorylation during mitosis in humans, the results were inconclusive about the effects of Aurora B-mediated phosphorylation of human Cyclin B1 during anaphase. A more comprehensive and systematic study of these and other putative phosphorylated residues, combined with tools that allow the expression of the mutants in a temporally controlled manner, will be required to establish a conserved mechanistic role of Cyclin B1 phosphorylation during anaphase by Aurora B. For these reasons, we decided to tone down the role of Aurora B on Cyclin B1 phosphorylation and removed all the data on Cyclin B1 phosphorylation by Aurora B from the revised manuscript and will investigate this further in future works.

Other points:1) The way the Aurora B and Subito experiments are presented and interpreted are confusing. The model proposed in Figure 10 predicts that a lack of Aurora B at the spindle midzone would accelerate cyclin B degradation due to the absence of a phosphorylation gradient that spatially restricts cyclin B1 on the spindle midzone. However, in Figure 8, and Figure 7—figure supplement 1, cyclin B1 degradation is delayed when Aurora B is not on the midzone, and anaphase duration prolonged. This could be due to the fact that after Subito knock-down, Aurora B is retained much longer on the chromatin (Figure 1–figure supplement 3f), similar to what happens after Aurora B overexpression (Figure 7—figure supplement 1). Having Aurora B (and cyclin B/cdk1) at the midzone allows the formation of a phosphorylation gradient (due to counteracting activities of PP2A/PP1). Movement of chromosomes away from the heart of the gradient would allow exit. When Aurora B (and cyclin B? – this is unclear) is retained on the chromatin, this spatial separation is obviously not possible. Moreover, it might also explain why degradation of cyclin B at centrosomes is delayed under these conditions as more Aurora B is in closer proximity of these structures. Related to this: does cyclin B colocalize with Aurora B on the chromatin after Subito depletion? Where do the cyclin B 5E and 5A mutants localize (apart from the centrosomes)? Potential localization on chromatin is masked by the H2B-GFP.

We would like to emphasize that neither WT nor mutant Cyclin B1 localize on chromatin after Subito depletion. This is clearly shown in Figure 8 for the WT Cyclin B1.

2) It looks as if depletion of Mklp2 affects cyclin B localization on the mitotic spindle before anaphase. Is this consistent? If so, doesn't this impact the cyclin B-GFP intensity measurements?

From our live-cell recordings we did not have any perception that Cyclin B1 localization on the mitotic spindle was affected before anaphase after Mklp2 depletion. In any case, even if it did, this would not affect our measurements of relative Cyclin B1 fluorescence decay, as the initial values between control and Mklp2-depleted cells are always normalized at 1 min prior to anaphase.

3) The lack of effect of acute PP1/PP2A inhibition on cyclin B1 degradation (Figure 9G) is difficult to reconcile with a model in which a phosphorylation gradient concentrates cyclin B/cdk1 activity at the spindle midzone (Figure 10), and with the observation that a cyclin B 5E mutant is more stable and delaying mitotic exit (Figure 9C,D). The latter would suggest that cyclin B needs to be dephosphorylated for timely degradation.A phosphorylation event on a protein substrate can be eliminated either by dephosphorylation or substrate degradation. Our data supports a model in which local activity of Aurora B at the spindle midzone stabilizes Cyclin B1, possibly by direct phosphorylation. Since PP1/PP2A inhibition at anaphase onset did not show significant differences in Cyclin B1 half-life (at least the centrosomal-associated pool measured in these assays; see new Figure 10—figure supplement 1C) and, on the contrary, Fzr depletion showed a significant delay in Cyclin B1 degradation (new Figure 5f, g h) our interpretation is that removal of Cyclin B1 phosphorylation is achieved by proteolysis and not by dephosphorylation. In our view, this does not exclude the critical role played by PP1/PP2A in the generation of a phosphorylation gradient that might operate to locally control Cyclin B levels at the spindle midzone.4) All referees had concerns with Figure 4:What is happening to the spindle and chromosomes after MG132 addition and subsequent Cdk1 inhibition in U2OS cells. It looks as if cytokinesis (abscission unclear) is taking place and that in these daughter cells the half spindle (00:10) is forming a new 'metaphase' spindle over time on which the separated sisters try to align (01:00 and 01:30). After Cdk1 inhibition, both spindles undergo an aberrant anaphase/exit. Can this phenotype be regarded as a delay in anaphase, or should it be considered as a reversion into prometaphase? In S2 cells, the anaphase delay is much more clear, because cytokinesis does not appear to take place and the two half spindles rejoin after Cdk1 inhibition.If cells only enter anaphase when Cdk1 activity has fallen below the required threshold to maintain metaphase, then how can cells arrested in anaphase by MG132 return to a metaphase-like state? Presumably this state results from blocking degradation events (cyclin B or other targets) normally contributing to disassembly of the spindle poles during anaphase. (And it implies that the central spindle – in contrast – disassembles under these conditions). Without further justification of the 'metaphase' label (e.g. restoration of Cdk activity), it should be avoided.Proteasomal inhibition in anaphase blocks mitotic exit. The authors interpret these results in an "anaphase arrest" This is true as far as cohesion cleavage is concerned. However, the spindle appears to reform and from a cytoskeletal and cell cycle control perspective it looks more as if the cells were simply being pushed back into a mitotic state. This experiment shows that if Cdk1 activity is maintained above a threshold mitotic exit is blocked, most likely because phosphatases fail to reactivate. The particular experimental set up (i.e. adding a proteasome inhibitor following anaphase onset) is novel; however, the results are not particularly surprising, given for example the results shown by Potapova et al. that even reactivation of Cdk1 in late telophase reactivates the mitotic state.

The reviewers are correct in their perception that cytokinesis is starting to take place after MG132 addition *at anaphase onset*, however abscission is not completed and the two ‘daughters’ remain connected by a thin cytoplasmic bridge. A higher contrast snapshot of the relevant frame in new Figure 4A is now provided to clarify this issue. Subsequent Cdk1 inhibition *during anaphase* induced complete chromosome decondensation and spindle disassembly *in a common cytoplasm* that attempts to complete furrowing (another indication that the two ‘daughters’ were indeed connected by a cytoplasmic bridge), three hallmarks of mitotic exit. The reviewers are also correct in their observations that cells attempt to revert from their anaphase state by forming a new ‘prometaphase-like’ spindle, but only after a significant delay (approximately 1 hour) in an ‘anaphase-like’ state. We also note that each individual sister chromatid set forms their own bipolar spindle, in which most single sister chromatids cannot establish a stable position at the equator due to lack of amphitelic attachments (some do, likely due to merotelic attachments, as shown previously by Khodjakov et al., 1997). This mitotic reversal has indeed been previously observed by Gorbsky and colleagues (Potapova et al., 2006), which we cite, after treating cells with MG132 followed by Cdk1 inhibition *in metaphase* followed by Cdk1 inhibitor washout in anaphase. However, in this case, sister chromatids never fully separated (cut phenotype) and only one spindle formed. The differences between our studies are now clarified in the main text and terminology has been improved to avoid ambiguity (e.g. ‘anaphase arrest’ was replaced by ‘prevented mitotic exit’ or ‘anaphase-like state’). The observation that Cdk1 inhibition after preventing proteasome activity during anaphase resumed mitotic exit, excludes the possibility that the observed delay in mitotic exit after proteasome inhibition was caused by phosphatases that failed to reactivate. This is discussed in the main text.

The movies and images do not really support the notion of accelerated mitotic exit by AurB inhibition. Likewise, in the legend for Figure 4C the authors state: Note that for MG132 and MG132+Aurora B inhibition anaphase duration corresponds to the total duration of the movie as most cells do not exit mitosis until the end of the acquisition. I am not sure how in this case accelerated mitotic exit following AurB inhibition can be supported by the data? Also, it is not clear what the grey, green, purple colour code means.We based our conclusion on the quantification of the duration between anaphase onset and mitotic exit in MG132-treated cells, with or without Aurora B inhibition. On average, this duration was shorter in cells after Aurora B inhibition, inclusively now some cells exited mitosis before the total duration of the movies, hence our statement. Yet, this is still much longer then Cdk1 inhibition. This is now clarified in the main text and figure legend, and the color code better explained also in the figure legend.5) The videos are not referred to in the text.This was our mistake, we apologize for the inconvenience. The movies are now referred in the main text.6) Degradation analyses of cyclin B1 usually show that cells enter anaphase with some protein remaining (in contrast to some figures in the original Clute & Pines 1999 study, which is cited to justify the statement at bottom of p8 that cyclin B1 is 'undetectable' at anaphase entry). The thresholding of anaphase entry underpins the phenomenon of mitotic slippage and has been explored at length in more recent literature that is not cited in this study. For example, a study from Stephen Taylor's lab (Gascoigne, 2009) measured a Cyclin B1-GFP threshold for anaphase corresponding to 32% of metaphase levels.This is an important point that escaped our attention. We thank the reviewers for this remark and we now cite and discuss the work by Taylor and colleagues in the main text.7) The phrase 'mitotic exit' needs to be used more carefully. It is conventionally used to describe the collective events that occur after anaphase onset and the authors sometimes use the phrase in this sense but in some places they use it to describe the point at which cells have completed these events (e.g. subsection “Cyclin B1 continues to be degraded during anaphase and its disappearance is a strong predictor of mitotic exit in metazoans” 'complete Cyclin B1 disappearance from centrosomes during anaphase strongly correlated with mitotic exit').We agree with this point, but we think that the conventional use of the term ‘mitotic exit’ has been misleading. In our view, mitotic exit is not the collective series of events that occur after anaphase onset, but the irreversible transition from anaphase to telophase. This can be recognized by spindle disassembly, chromosome decondensation and complete nuclear envelope reassembly, but excludes for example the events occurring *during* anaphase, such as poleward chromosome motion and spindle elongation. This view is now clarified in the main text.8) No mention is made of the likely presence of cyclin B3 in anaphase cells. B3 is degraded later than B1, and could contribute to Cdk1-dependent delay in mitotic exit.This is an excellent point. Although we did mention in the Introduction section that “expression of non-degradable Cyclin B1 (and B3 in Drosophila) mutants arrested cells in anaphase”, now we investigated the importance of Cyclin B3 degradation during anaphase by monitoring mitotic progression in live S2 cells expressing non-degradable Cyclin B3. We found that, similar to expression of non-degradable Cyclin B1, S2 cells expressing non-degradable Cyclin B3 also arrested in anaphase with spindle elongation problems. Thus, both Cyclin B1 and B3 account for residual Cdk1 activity during anaphase in Drosophila S2 cells. These data is now shown in new Figure 2—figure supplement 1C-E and discussed in the main text. We have also used the terminology ‘B-type Cyclins’ to reflect these results.9) Figure 5 shows a very small effect of FZR depletion on cyclin B1 degradation at anaphase (especially in (f)) suggesting that Cdh1 may only play a minor role. The authors argue that prolonged Cdc20 activity can compensate for lack of Cdh1 under these conditions, by targeting cyclin B1 through its D-box (whereas Cdh1 targets it through KEN). These arguments do not properly reflect the latest thinking on patterns of APC/C-degron recognition. Cryo-EM structures published over the recent years have established a paradigm of multivalent APC/C-substrate interaction through two or more degrons and Cdc20 and Cdh1 have almost identical KEN and D-box receptor sites. It would be safer to conclude that Drosophila Cyclin B1 degradation in anaphase is mediated through both KEN and D-box, and can be carried out by either Cdc20 or Cdh1. This could be tested by the combination of KEN-mt/APCin.The authors should note that APCin cannot be expected to inhibit a KEN-box dependent event, since it is a D-box competitor, so the discussions at top of p6 and middle p10 are incorrect – the additive effect of APCin and proTAME on anaphase arrest does not support a role for Cdh1 in anaphase cyclin B1 degradation. It supports that additive interactions of Cdc20/Cdh1 and substrate degrons are required for efficient ubiquitination of substrates.We agree with this point and have now re-interpreted the conclusions from these experiments alluding to the recent Cryo-EM structures of APC/C. This re-interpretation is also supported by additional experiments where we combined acute Apcin treatment at anaphase onset with our KENbox mutant and found a synergistic effect. This result also suggests the presence of additional recognition sites that mediate Cyclin B1 degradation through APC/C during anaphase. These new data are now included in new Figure 5—figure supplement 1A-D) and discussed in the main text. We thank the reviewers for suggesting this experiment.10) Effects on cyclin B1-GFP degradation look very small in regular degradation plots, and are more convincingly presented as half-life measurements (e.g. Figure 8E, Figure 1—figure supplement 3D). The degradation curve for OA treatment (Figure 9E) also appears to show a very small difference in rate between treated and untreated cells; these data should likewise be presented as half life measurements to provide same level of proof as in other figures.We have now measured Cyclin B1-GFP half-life at centrosomes with and without OA but the results revealed no statistically significant differences.11) As far as I am aware, 'anaphase arrest' (Figure 5J) has never been described as a consequence of FZR depletion in mammalian cells. Could additional markers be used to show that this is really an anaphase arrest (as opposed to death in anaphase)?This is another important point. In fact, the term ‘anaphase arrest’ is not appropriate because we are looking here at fixed cells. Nevertheless, individual sister chromatids that remain condensed are clearly visible by DAPI staining and the inclusion of a kinetochore marker (CENP-C) allowed us to unequivocally classify those cells as ‘anaphases’ due to the presence of individual kinetochores attached to microtubules from two separated half-spindles, as typically observed in anaphase cells expressing non-degradable Cyclin B1 or treated with MG132. We have now clarified this point in the main text.12) To my knowledge Cdk1 inhibition after anaphase onset has not been analysed before and it makes sense that this causes a faster reformation of the NE. This can be explained by the Cundell et al., papers due to premature Greatwall inactivation and PP2A:B55 reactivation. These papers should be cited here.

We now cite the works of Cundell at el. in our Discussion. We appreciate the suggestion.

[Editors' note: further revisions were requested prior to acceptance, as described below.]

The revised paper has addressed the substantive criticisms of the reviewers but before publication you will need to modify the text to take account of the following points, and in particular to tone down your conclusions for human cells:1) Subsection “Cyclin B1 degradation during anaphase is mediated by APC/C^Cdh1^: Glotzer et al., 1991 is not the correct reference for Cyclin B1 degradation under the control of the SAC. The correct reference is Clute and Pines, 1999.

We have corrected the references to reflect each specific contribution. It now reads as follows:

“APC/CCdc20 is thought to regulate the metaphase-anaphase transition by targeting Cyclin B1 for degradation through recognition of a D-box (Glotzer et al., 1991), under control of the SAC (Clute and Pines, 1999).”

2) Subsection “Cyclin B1 degradation during anaphase is mediated by APC/C^Cdh1^: The caveat to the APCin/TAME experiment is that both inhibitors might be needed because they are relatively inefficient, not that Cdc20 and Cdh1 are additive in anaphase.

We now discuss this possibility. It now reads as follows:

“Only the simultaneous addition of both drugs at anaphase onset blocked the continuous degradation of Cyclin B1 during anaphase and prevented mitotic exit for several hours, similar to proteasome inhibition (Figure 6a-f). Although relative inefficiency of these inhibitors cannot be ruled out, it is conceivable that additive interactions of Cdc20/Cdh1 and Cyclin B1 degrons are required for efficient Cyclin B1 ubiquitination and subsequent degradation by the proteasome. Taken together, these results suggest that *Drosophila* and human Cyclin B1 degradation during anaphase is mediated through the D-box and KEN-box (when present), and can be carried out either by APC/CCdc20 or APC/CCdh1.”

3) Subsection “Cyclin B1-Cdk1 localization and activity during anaphase are spatially 224 regulated by Aurora B at the spindle midzone”: The caveat to Figure 7 is that the presence of Cdk1 does not imply the presence ofCyclin B1.

We have now re-written the section sub-title to specifically mention that Cyclin B1 was only detected in the spindle midzone in *Drosophila* cells. We have also significantly re-written the section to clarify the point raised. It now reads as follows:

“While we were unable to localize endogenously tagged Cyclin B1-Venus at the spindle midzone in human anaphase cells due to the poor signal/noise at this stage, direct localization of human Cdk1-GFP in live human HeLa cells revealed a clear spindle localization at the metaphase-anaphase transition, with a subsequent enrichment in the central spindle region between separating sister chromosomes throughout anaphase, and midbody during cytokinesis (Figure 7—figure supplement 3A,B). In agreement, Cdk1 was recently found enriched on isolated midbodies from human cells (Capalbo et al., 2019). These results suggest that Cyclin B1-Cdk1 is spatially regulated by Aurora B activity at the spindle midzone, at least in *Drosophila* cells.”

4) Discussion section: APC/C-Cdc20 degrades Cyclin B1 in metaphase not at the meta-and transition.

We have now revised this in the model of Figure 11 and corrected this point in the main text. It now reads as follows:

“According to this model, APC/CCdc20 mediates the initial degradation of Cyclin B1 during metaphase under SAC control. The consequent decrease in Cdk1 activity as cells enter anaphase…”

5) Discussion section: You have not shown that human Cyclin B1 persists at the spindle midzone, just Cdk1.

We have now re-written this section to reflect the point raised. It now reads as follows:

“Aurora B at the spindle midzone (counteracted by PP1/PP2A phosphatases on chromatin (Vagnarelli et al., 2011)) establishes a phosphorylation gradient that locally delays APC/CCdc20- and APC/CCdh1-mediated degradation of residual Cyclin B1 (and possibly B3) at the spindle midzone, at least in *Drosophila* cells. Localization experiments in human cells suggest that Cdk1 itself might be enriched at the spindle midzone. Consequently, as chromosomes separate and move away from the spindle midzone, Cdk1 activity decreases, allowing the PP1/PP2A-mediated dephosphorylation of Cdk1 and Aurora B substrates (e.g. Lamin B and Condensin I) necessary for mitotic exit.”